# The critical role of AMPK in driving Akt activation under stress, tumorigenesis and drug resistance

Fei Han[1,2], Chien-Feng Li[3,4], Zhen Cai[1,2], Xian Zhang (ID) [1,2], Guoxiang Jin[1,2], Wei-Na Zhang[1], Chuan Xu[1], Chi-Yun Wang[1,2], John Morrow[5], Shuxing Zhang[5], Dazhi Xu[2,6], Guihua Wang[1] & Hui-Kuan Lin[1,2,7,8]

PI3K/Akt signaling is activated in cancers and governs tumor initiation and progression, but how Akt is activated under diverse stresses is poorly understood. Here we identify AMPK as an essential regulator for Akt activation by various stresses. Surprisingly, AMPK is also activated by growth factor EGF through Ca2+/Calmodulin-dependent kinase and is essential for EGF-mediated Akt activation and biological functions. AMPK phosphorylates Skp2 at S256 and promotes the integrity and E3 ligase activity of Skp2 SCF complex leading to K63-linked ubiquitination and activation of Akt and subsequent oncogenic processes. Importantly, AMPK-mediated Skp2 S256 phosphorylation promotes breast cancer progression in mouse tumor models, correlates with Akt and AMPK activation in breast cancer patients, and predicts poor survival outcomes. Finally, targeting AMPK-mediated Skp2 S256 phosphorylation sensitizes cells to anti-EGF receptor targeted therapy. Our study sheds light on how stress and EGF induce Akt activation and new mechanisms for AMPK-mediated oncogenesis and drug resistance.

[1] Department of Cancer Biology, Wake Forest School of Medicine, Winston-Salem, NC 27157, USA. [2] Department of Molecular and Cellular Oncology, The University of Texas MD Anderson Cancer Center, Houston, TX 77030, USA. [3] Department of Pathology, Chi-Mei Foundational Medical Center, Tainan 710, Taiwan. [4] National Institute of Cancer Research, National Health Research Institutes, Tainan 704, Taiwan. [5] Department of Experimental Therapeutics, The University of Texas MD Anderson Cancer Center, Houston, TX 77030, USA. [6] Department of Gastric and Pancreatic Surgery, Sun Yat-Sen University Cancer Center, Guangzhou 510060, China. [7] Graduate Institute of Basic Medical Science, China Medical University, Taichung 404, Taiwan. [8] Department of Biotechnology, Asia University, Taichung 41354, Taiwan. Correspondence and requests for materials should be addressed to D.X. (email: xudzh@sysucc.org.cn) or to G.W. (email: ghwang18@163.com) or to H.-K.L. (email: hulin@wakehealth.edu)

PI3K/Akt signaling governs a variety of cellular functions such as proliferation, metabolism, cell survival, and migration critical for tumor initiation and progression[1]. Many growth factors and cytokines are known to activate PI3K/Akt through binding with their membrane receptor and activating receptor tyrosine kinases. Once PI3K is activated, it catalyzes the phosphorylation of PI(4,5)P2 to form PI(3,4,5)P3, which then recruits Akt to the cell plasma membrane[2]. Akt binds to PI(3,4,5)P3 phospholipid via its N-terminal PH domain, which is required for its recruitment to the cell plasma membrane[3,4]. Upon membrane recruitment, Akt is phosphorylated by PDK1 at Thr308 in the activation loop of the kinase domain, in turn leading to Akt activation. Full activation of Akt requires phosphorylation at Ser473 located in the regulatory domain by mTORC2. Once Akt is fully activated, it then phosphorylates numerous downstream effectors to orchestrate diverse biological processes important for tumorigenesis such as cell proliferation, survival, and metabolism[5].

While PI(3,4,5)P3 formation induced by PI3K is clearly critical for membrane recruitment and activation of Akt upon growth factor stimulation, recent studies have revealed that K63-linked ubiquitination of Akt induced by growth factors is also a prerequisite for these processes[6,7]. Interestingly, while diverse growth factors commonly induce K63-linked ubiquitnaiton of Akt to facilitate Akt membrane recruitment and activaiton, distinct E3 ubiquitin ligases are utilized by grwoth factors for K63-linked ubiquitnaiton of Akt. TRAF6 E3 ligase is selectively activated and ubiquitinates Akt in response to IGF-1 treatment, whereas Skp2 SCF E3 ligase is activated and responsible for K63-linked ubiquitination of Akt upon EGF stimulation[6,7]. Deficiency of TRAF6 or Skp2 impairs K63-linked ubiquitination, cell membrane localization and activation of Akt, resulting in tumor suppression in mouse tumor models[6,7]. However, how growth factors activate TRAF6 and Skp2 to promote Akt ubiquitination is largely unknown. Since Akt phosphorylation and activation are also induced by other extracellular and intracellular cues, whether K63-linked ubiquitination of Akt is generally induced and serves as a common mechanism for Akt phosphorylation and activation by these stimuli remains puzzling.

During solid tumor progression, tumor cells are often exposed to hypoxic environments because they are located away from blood vessels and thus have a limited oxygen supply. Although severe hypoxia usually leads to tumor necrosis, moderate hypoxia near the center of tumor promotes tumor angiogenesis, cancer cell survival, and stemness, thereby promoting cancer progression, metastasis, and drug resistance[8]. PI3K/Akt appears to be activated and is responsible for cancer cell survival under hypoxia in diverse cell types[9–11], although the underlying mechanism by which PI3K/Akt are activated is not well understood. Apart from hypoxia, other physiological and pathologic stresses, such as oxidative stress, glucose deprivation, ER stress, and DNA damage, are reported to induce Akt phosphorylation and activation[12,13], which may also help protect cancer cells from apoptosis under these stresses. However, the regulatory mechanism underlying Akt activation by these stresses remains elusive.

Lung cancer is a highly aggressive cancer type with poor prognosis, which is the leading cause of death worldwide with 5-year survival rate of less than 16%[14]. Among lung cancer subtypes, non-small cell lung cancer (NSCLC) represents the majority of lung cancer types, which composes around 80–85% of total lung cancer incidence. Chemotherapy and anti-EGFR targeted therapy agents are the first line treatment options for NSCLC. While patients respond to these treatments initially, resistance to these treatments soon develops, thereby leading to cancer recurrence and mortality[15]. While the resistant mechanisms are not yet well understood, the activation of PI3K/Akt pathways appears to contribute to this resistance[16]. Thus, understanding the upstream regulators orchestrating PI3K/Akt activation during cancer progression and resistance may offer new strategies for overcoming drug resistance in lung cancer treatment.

In this study, we aimed to unveil the novel player participating in Akt activation under diverse stresses. We discovered the stress kinase AMPK is critical for this process. AMPK phosphorylated Skp2 at S256 and induced K63-linked ubiquitination and activation of Akt under diverse stresses. Moreover, we also found the CaMKKβ/AMPK/Skp2 axis is important for EGF-induced Akt phosphorylation and activation. Our study not only explains how Skp2 SCF E3 ligase is activated under EGF to trigger Akt ubiquitination and activation, but also provides a possible resistant mechanism for anti-EGFR targeted therapy.

## Results

**AMPK is required for Akt phosphorylation and activation under various cellular stresses**. Akt (also known as PKB) is a serine-threonine kinase that plays important roles in cancer initiation and progression by regulating diverse biological processes such as cell growth and survival. In addition to growth factors, Akt signaling is also activated by diverse stresses including hypoxia, metabolic stress, and oxidative stress. However, the underlying mechanism for Akt activation under these stresses is poorly understood. We hypothesized that a common mechanism may be responsible for Akt activation by these stresses. To identify possible kinases other than PI3K responsible for stress-induced Akt activation, we first applied a kinase inhibitor library, which includes a panel of inhibitors against many key kinases of multiple cellular processes (Supplementary Table 1), to examine Akt activation in HEK293 cells under hypoxia challenge. We monitored Akt S473 phosphorylation (pAkt S473) as a readout for Akt activation in order to minimize experimental variations because it has a stronger and more stable signal. As expected, inhibitors blocking activation of PI3K, PDK1, PIM, and Src, known to be involved in Akt activation, all inhibited Akt phosphorylation upon hypoxia (Fig. 1a)[17,18] Moreover, inhibitors that target cell membrane receptors including IGF-1R, VEGFR, PDGFR, FGFR, and c-MET blocked Akt phosphorylation upon hypoxia (Fig. 1a). This suggests that hypoxia-induced Akt activation likely engages multiple cell membrane growth factor receptors. In support of this notion, hypoxia has been shown to induce PDGFR activation to mediate Akt activation[19]. Surprisingly, we found that compound C, an inhibitor of AMPK, also suppressed Akt phosphorylation (Fig. 1a). We also confirmed this result with compound C in MDA-MB-231 cells (Supplementary Fig. 1a), suggesting that AMPK may also be required for Akt activation upon hypoxia stimulation.

AMPK, a stress kinase that serves as an energy sensor to monitor the ATP/ADP ratio, is activated under various stresses including hypoxia[20]. It shares many similar downstream targets with Akt like tumor suppressor TSC2, but often has an opposite effect compared with Akt[21]. While AMPK is generally viewed as a tumor suppressor partly by phosphorylating TSC2 and Raptor, leading to inhibiting mTOR signaling, it can also display tumor-promoting activity by regulating NADPH and ROS levels in breast cancer[22,23]. It appears that AMPK exhibits dual functions either in tumor promoting or tumor suppressive activities, which may be determined by distinct cellular and/or tissue contexts. To further corroborate the notion that AMPK is critical for hypoxia-induced Akt activation, we took genetic approaches by generating AMPK stable knockdown cells using lentiviral shRNAs in MDA-MB-231 cells and observed a decreased level of phosphorylation

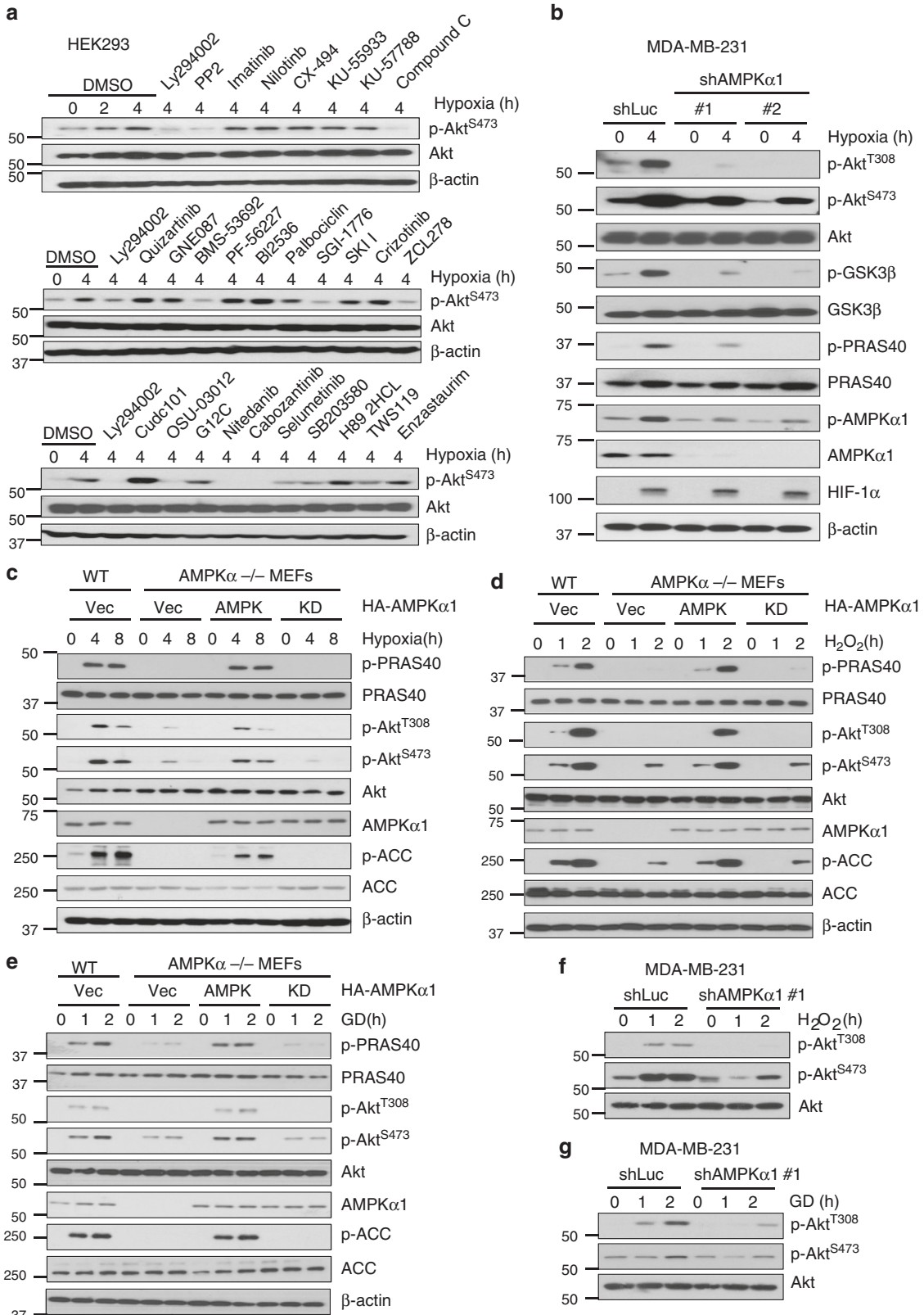

**Fig. 1** AMPK is required for Akt phosphorylation and activation under various cellular stresses. **a** Screening of kinases in the hypoxia-induced Akt phosphorylation using kinase inhibitor library. Immunoblotting of HEK293 cells pretreated with indicated inhibitors at 10 μM for 2 h and challenged with 1% O2 for another 4 h. **b** Immunoblotting of control (shLuc) and AMPKα1 knockdown MDA-MB-231 cells (#1, #2) treated with 1% O2 for 4 h. **c**–**e** Immunoblotting of WT and AMPKα1/α2 double knockout (AMPKα$^{-/-}$) restored with vector, WT or kinase dead AMPKα under hypoxia (1% O2), 100 μM H2O2 or glucose deprivation for indicated time. In each restoration experiment, 5 μg of the plasmid was transfected into indicated cell lines. **f**, **g** Immunoblotting of with control (shLuc) and AMPKα1 knockdown MDA-MB-231 cells under H2O2 (100 μM) or glucose deprivation for indicated time

of Akt at T308 and S473 and its downstream, GSK3β and PRAS40 (Fig. 1b). Strikingly, restoration of AMPK expression in AMPK knockdown cells rescued Akt phosphorylation and activation upon hypoxia treatment (Supplementary Fig. 1b). Similarly, hypoxia-induced Akt activation was also impaired in AMPKα1/α2 double knockout (AMPKα[-/-]) mouse embryonic fibroblasts (MEFs) (Supplementary Fig. 1c). Restoration of AMPK, but not kinase-dead AMPKα, rescued the defect in Akt phosphorylation in AMPKα[-/-] MEFs upon hypoxia stimulation (Fig. 1c). To extend our findings in other cancer cell lines, we also generated AMPK knockdown in HCT116 and MCF7 cells and found that hypoxia-induced Akt phosphorylation was also impaired upon AMPK knockdown (Supplementary Fig. 1d, e). These data suggest that AMPK is required for hypoxia-induced Akt activation.

Apart from hypoxia, other stresses such as glucose deprivation and oxidative stress also induce Akt activation through mechanisms that are not yet clear[24,25]. We determined whether AMPK is globally required for Akt activation under other stressed conditions. $H_2O_2$ and glucose deprivation-induced Akt phosphorylation was also impaired in MDA-MB-231 cells and AMPKα[-/-] MEFs, while restoration of AMPK, but not kinase-dead AMPKα, rescued the defect in Akt phosphorylation in AMPKα[-/-] MEFs under these stressed conditions (Fig. 1d–g). Similar results were also obtained in HCT116, BT474, and MDA-MB-361 cancer cells, where Akt phosphorylation upon glucose deprivation was impaired in AMPK knockdown cells (Supplementary Fig. 1f-g). Moreover, impaired Akt phosphorylation induced by $H_2O_2$ or glucose deprivation was also observed in Compound C-treated MDA-MB-231 cells (Supplementary Fig. 1i-j). Thus, AMPK is globally required for Akt activation under diverse stresses.

**Ca2+/CaMKKβ-AMPK signaling is critical for EGF-induced Akt activation.** Akt signaling is known to be a major pathway for maintaining cell proliferation and survival in response to various growth factors, whereas AMPK is involved in cell survival under energy stress. Whether or not AMPK is involved in EGF signaling and its biological functions remains puzzling. Having shown that AMPK is essential for stress-induced Akt activation, we then examined whether AMPK is also involved in Akt phosphorylation and activation by diverse growth factors. Interestingly, AMPK was also required for Akt phosphorylation and activation upon EGF treatment (Fig. 2a), although it was not statistically significant for Akt phosphorylation and activation by IGF-1 and PDGF in MDA-MB-231 cells (Supplementary Fig. 2a, b). Similarly, Akt activation under EGF treatment was also impaired in AMPKα[-/-] MEFs and restoration of AMPK, but not kinase-dead AMPKα, rescued the defect in Akt phosphorylation in AMPKα[-/-] MEFs upon EGF treatment (Fig. 2b). Similar results were also obtained in AMPK knockdown HCT116, BT474 and MDA-MB-361 cells and compound C-treated MDA-MB-231 cells (Supplementary Fig. 2c, d). We observed that EGF could induce AMPK-dependent ACC phosphorylation, indicative of AMPK activation by EGF (Fig. 2a, b). However, EGF-induced EGFR and its downstream ERK phosphorylation was not affected upon AMPK deficiency (Fig. 2a). These results demonstrate that AMPK is activated and essential for Akt activation in response to growth factor EGF and diverse stresses.

The finding that AMPK is activated and crucial for Akt activation by EGF is quite surprising, given that the stress kinase AMPK is generally regarded as a cellular sensor that is activated when ATP levels are low. LKB1 and CaMKKβ are two major upstream kinases for AMPK activation[26,27]. To gain insight into how AMPK is activated by EGF, we examined whether EGF-induced activation of AMPK and Akt acts through one of these two known kinases. Activation of AMPK and Akt by EGF was impaired upon both CaMKKβ knockdown and CaMKKβ inhibitor STO609 treatment (Fig. 2c, d). Restoration of CaMKKβ rescued the defect in Akt phosphorylation in CaMKKβ knockdown MDA-MB-231 cells upon EGF treatment (Fig. 2e). Similar to that in AMPK deficient cells, EGFR and ERK phosphorylation in either CaMKKb deficient or CaMKKb addback cells remained unchanged (Fig. 2c, e). However, EGF-induced AMPK and Akt activation was not affected by LKB1 knockdown (Supplementary Fig. 2e).

CaMKKβ is activated when intracellular $Ca^{2+}$ level arises. To explore how EGF acts through CaMKKβ to drive AMPK and Akt activation, we traced intracellular $Ca^{2+}$ level following EGF treatment by Fura-2 AM, a permeable calcium indicator. The intracellular $Ca^{2+}$ level was rapidly elevated upon EGF treatment; however, blocking the increase of the $Ca^{2+}$ level by an intracellular calcium chelator BAPTA abrogated EGF-induced AMPK and Akt activation (Fig. 2f and Supplementary Fig. 2f). However, PDGF failed to elevate intracellular $Ca^{2+}$ level in MDA-MB-231 cells (Supplementary Fig. 2g), consistent with the fact that AMPK is not involved in Akt activation upon PDGF treatment (Supplementary Fig. 2h). These data suggest that EGF induces intracellular $Ca^{2+}$ increase to activate CaMKKβ-AMPK signaling, in turn leading to Akt activation.

Since mTORC1 is a downstream effector of Akt under growth factors and AMPK also drives mTORC1 inactivation through phosphorylation and activation of TSC2 and Raptor, we investigated the role of mTORC1 in AMPK dependent Akt activation under diverse stresses and EGF. Control and AMPK knockdown MDA-MB-231 cells were treated with Rapamycin under hypoxia, glucose deprivation, $H_2O_2$ or EGF. While phosphorylation of mTOR downstream effectors, S6K and Grb10, was induced under these conditions, AMPK knockdown has subtle impact on phosphorylation of S6K and Grb10 in our cell models. However, AMPK knockdown consistently impaired Akt phosphorylation even under the treatment of mTORC1 inhibitor rapamycin (Supplementary Fig. 3a-d). As a control, we found that mTORC1 inhibitor rapamycin inhibited S6K and Grb10 phosphorylation. These data suggest that AMPK dependent Akt activation under hypoxia, glucose deprivation, H2O2 and EGF partly acts through an mTORC1-independent mechanism.

**Skp2 and AMPK are crucial for stress-induced Akt ubiquitination and activation.** Skp2 SCF complex is an E3 ligase that mediates Akt ubiquitination and activation upon EGF stimulation, leading to the promotion of breast cancer development[6]. However, it is unclear whether Akt ubiquitination is also a common event for Akt phosphorylation and activation under diverse stresses. To answer this question, we examined whether Akt ubiquitination is also induced by stresses. As shown in Fig. 3a, Endogenous Akt ubiquitination was induced by hypoxia. Since Lys (K48)-linked ubiquitination leads to proteasome-dependent degradation while K63-linked ubiquitination usually promotes signaling activation, we dissected which type of Akt ubiquitination was induced under hypoxia. The majority of Akt ubiquitination induced by hypoxia involved K63-linked, but not K48-linked, polyubiquitin chains (Fig. 3b). To examine whether AMPK is required for endogenous K63-linked ubiquitination of Akt, we performed the in vivo ubiquitination assay by using the antibody specifically recognizing K63-linked ubiquitination and demonstrated that hypoxia-induced endogenous K63-linked ubiquitination of Akt was impaired in AMPK knockdown cells (Fig. 3c). To further dissect the role of AMPK in Akt

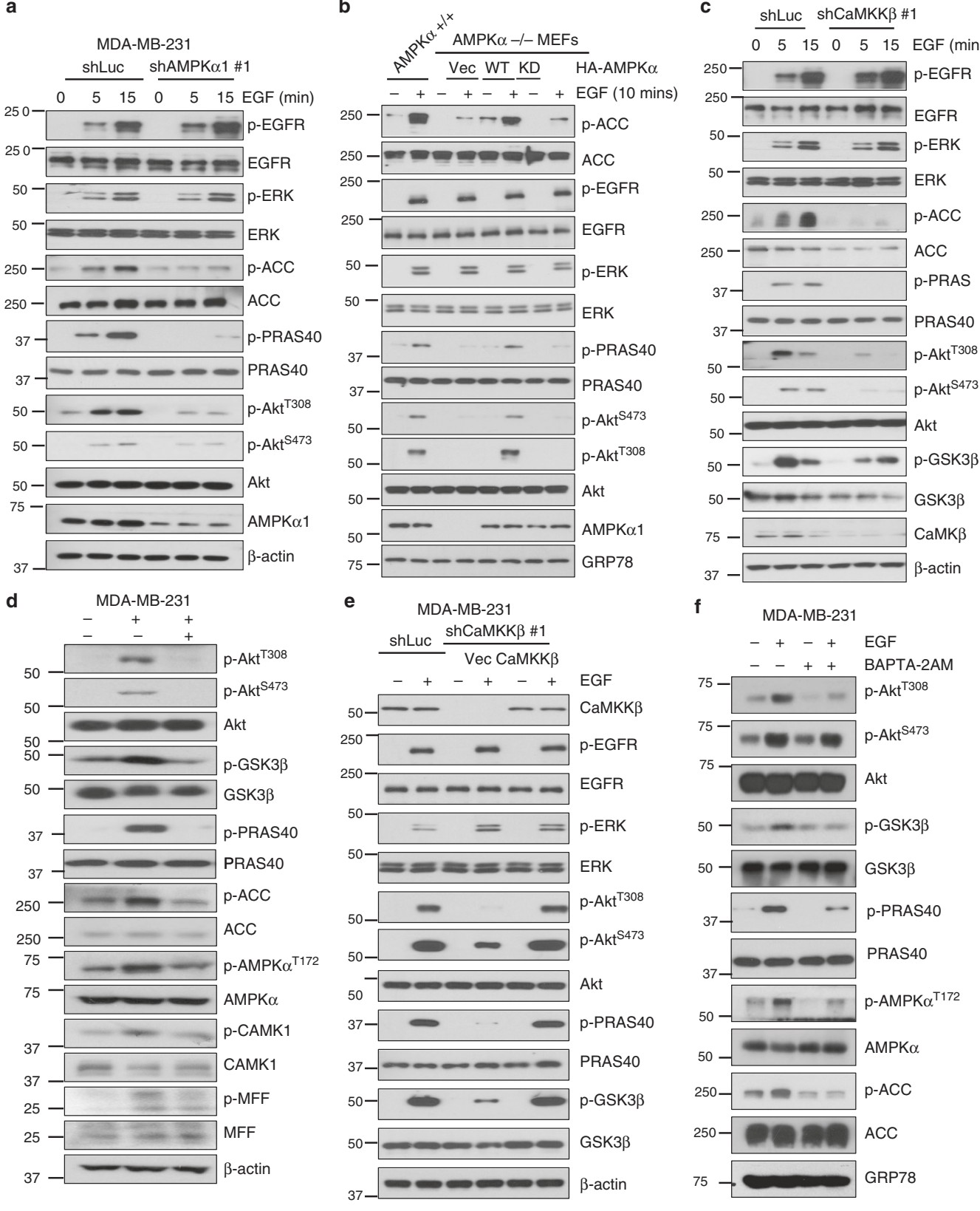

**Fig. 2** $Ca^{2+}$/CaMKKβ-AMPK signaling is critical for EGF-induced Akt activation. **a** Immunoblotting of control (shLuc) and AMPKα1 knockdown (#1) MDA-MB-231 cells starved for 16 h and treated with 50 ng/ml EGF for 5 and 15 min. **b** Immunoblotting of $AMPKα^{+/+}$ and $AMPKα^{-/-}$ MEFs restored with vector, WT and kinase dead AMPKα starved for 16 h and treated with 50 ng/ml EGF for 10 min. In each restoration experiment, 5 μg of the plasmid was transfected into indicated cell lines. **c** Immunoblotting of control (shLuc) and CaMKKβ knockdown (#1) MDA-MB-231 cells serum starved and treated with EGF for 5 and 15 min. **d** Immunoblotting of MDA-MB-231 cells treated with EGF (50 ng/ml) for 5 min and STO609 (10 μM) for 5 min. **e** Immunoblotting of control (shLuc), CaMKKβ knockdown (#1) restored with vector and CaMKKβ MDA-MB-231 cells serum starved and treated with EGF for 10 min. In each restoration experiment, 5 μg of the plasmid was transfected into indicated cell lines. **f** Immunoblotting of MDA-MB-231 cells pretreated with BAPTA-2AM (10 μM) for 15 min and treated with EGF (50 ng/ml) for another 5 min

ubiquitination under stresses, we restored AMPK in $AMPKα^{-/-}$ MEFs and assessed Akt ubiquitination under stresses. We found Akt ubiquitination was impaired in $AMPKα^{-/-}$ MEFs, but AMPK restoration rescued the defect in Akt ubiquitination in $AMPKα^{-/-}$ MEFs upon hypoxia, glucose deprivation or $H_2O_2$ treatment (Fig. 3d–f). Similar to AMPK, Skp2 was not only required for EGF-mediated Akt ubiquitination and activation, but also for hypoxia-induced and $H_2O_2$-induced K63-linked ubiquitination and activation of Akt (Fig. 3g–i). Thus, K63-linked ubiquitination of Akt is commonly induced by not only growth factor EGF, but also by distinct stresses, such as hypoxia and $H_2O_2$, in a Skp2-dependent and AMPK-dependent manner.

**Skp2 S256 phosphorylation by AMPK promotes Skp2 SCF complex formation and its E3 ligase activation.** Given that the Skp2 SCF complex is an E3 ligase responsible for Akt ubiquitination and activation in response to EGF and diverse stresses, we speculated that AMPK may serve as an upstream kinase responsible for activation of the Skp2 SCF complex, thereby leading to Akt ubiquitination and activation under various stimuli. To test this hypothesis, we examined whether Skp2 is a substrate of AMPK by performing an in vitro kinase assay and found that active AMPK, but not kinase-dead AMPK, purified from 293 cells expressing active or kinase-dead AMPK directly phosphorylates Skp2 in vitro (Supplementary Fig. 4a). We also detected endogenous binding between AMPK and Skp2 in vivo (Supplementary Fig. 4b). To map the Skp2 phosphorylation region, we transfected cells with active AMPK and different truncated forms of Skp2 and found that AMPK phosphorylated the Skp2 ΔN mutant (aa 94–436), but not the N-terminal fragment (aa 1–200) (Fig. 4a). Interestingly, the C-terminal region (aa 200–436) of Skp2 contains four putative phosphorylation sites (S256, S367, S382 and S383) for AMPK based on the prediction of the Scansite under low stringency (Supplementary Fig. 4c). We then mutated these putative residues from serine (S) to alanine (A) and found that the Skp2 S256A mutant, but not other Skp2 mutants, displayed reduced Skp2 phosphorylation by active AMPK (Supplementary Fig. 4d), indicative of Skp2 S256 phosphorylation by AMPK. To validate this notion, we performed the mass spectrometry analysis in Skp2 immunoprecipitated samples isolated from 293 cells expressing vector or active AMPK. Notably, we detected Skp2 S256 phosphorylation in cells expressing active AMPK, but not in cells without expressing active AMPK (Supplementary Fig. 4e). Collectively, these data suggest that AMPK phosphorylates Skp2 at S256, which is highly conserved among species and shared similar consensus motif with other known AMPK substrates (Fig. 4b and Supplementary Fig. 4f).

To better understand the importance of AMPK-mediated Skp2 S256 phosphorylation, we generated a Skp2 phospho-specific antibody recognizing Skp2 S256 phosphorylation and used it for in vivo and in vitro kinase assays. AMPK-mediated Skp2 S256 phosphorylation could be detected in wild-type (WT) Skp2 transfected cells but not in cells transfected with Skp2 S256A

(Fig. 4c). In vitro kinase assay revealed that active AMPK, but not kinase-dead AMPK, purified from HEK293 cells phosphorylated Skp2 at S256, while Skp2 S256A could not be phosphorylated by active AMPK (Fig. 4d, e). To further solidify the notion that AMPK is a direct kinase for Skp2, we have repeated the in vitro kinase assay using recombinant GST-Skp2 purified from E. coli and active AMPKα1β1γ1 complex purified from Sf9 cells. We demonstrated that Skp2 S256 phosphorylation was triggered by AMPKα1β1γ1, but such phosphorylation was markedly inhibited by the AMPK inhibitor, compound C (Fig. 4f). To investigate the physiological relevance of Skp2 S256 phosphorylation, we detected Skp2 S256 phosphorylation under diverse stimuli and found that Skp2 S256 phosphorylation was induced by EGF, hypoxia, glucose deprivation or $H_2O_2$-induced oxidative stress (Fig. 4g). Remarkably, AMPK knockdown impaired Skp2 S256 phosphorylation upon these stimuli (Fig. 4g). To elucidate whether Skp2 S256 phosphorylation is regulated by AMPK, we restored AMPKα into AMPKα null MEFs and examined Skp2 S256 phosphorylation under stresses. Add-back of WT AMPK rescued Skp2 S256 phosphorylation under EGF, glucose deprivation or hypoxia treatment (Supplementary Fig. 5a). To investigate whether Akt signaling is regulated by AMPK through Skp2 S256 phosphorylation, we restored Skp2 S256A mutant or Skp2 WT into Skp2 null MEFs and assessed Akt signaling under hypoxia stimulation. Notably, Skp2 WT, but not Skp2 S256A, rescued the defect in Akt phosphorylation at T308 and PRAS40 phosphorylation under the EGF, hypoxia, glucose deprivation or $H_2O_2$ treatment (Supplementary Fig. 5b-e). We also introduced HA-AMPKα1 along with Akt WT or Akt ubiquitination deficient mutant (HA-Akt K8R) into AMPKα null MEFs and examined Akt signaling under the treatment of EFG, hypoxia, glucose deprivation or $H_2O_2$. While restoration of AMPK and WT Akt could rescue defects in Akt and PRAS40 phosphorylation, HA-Akt K8R failed to do so (Supplementary Fig. 5f-i), indicating that AMPK regulates Akt signaling through Akt ubiquitination. Collectively, these results suggest that AMPK is a direct kinase of Skp2 and induces Skp2 S256 phosphorylation in response to distinct stresses and EGF.

Since Skp2 SCF is an E3 ligase that triggers ubiquitination of its substrates, we next examined the role of AMPK-mediated Skp2 S256 phosphorylation in Skp2 SCF E3 ligase activity. WT Skp2 or phospho-mimic Skp2 mutant (Skp2 S256D) markedly promoted Akt ubiquitination, but Skp2 S256A mutant reduced this activity (Fig. 4h). Similarly, the Skp2 S256A mutant also failed to promote ubiquitination of p27, another substrate of Skp2 SCF (Supplementary Fig. 5j). Thus, Skp2 S256 phosphorylation is critical for Skp2 SCF E3 ligase activity. Given that Skp2 functions as an F-box protein that interacts with Skp1 and Cullin1 to form an SCF complex and that the complex integrity is important for its E3 ligase activity[28], we then tested whether Skp2 S256 phosphorylation is necessary for Skp2 SCF complex formation. Active AMPK, but not kinase-dead AMPK, promoted the interaction of Skp2 with Cullin1 (Cul1) and slightly enhanced the interaction between Skp2 and Skp1 (Supplementary Fig. 5k). Consistent with this notion, Skp2 and Skp2 S256D readily formed a complex

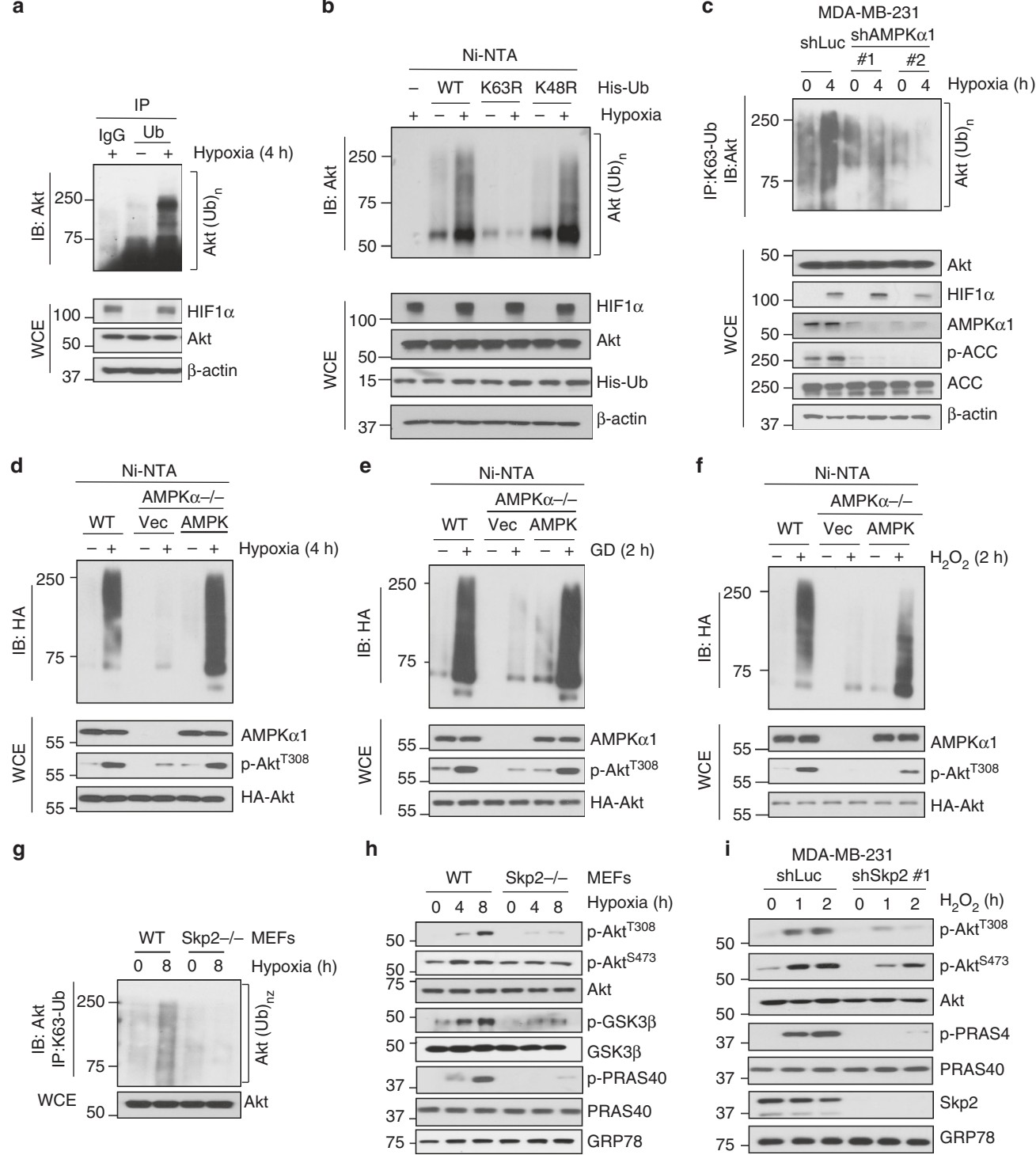

**Fig. 3** Skp2 and AMPK are crucial for stress-induced Akt ubiquitination and activation. **a** HEK293 cells in normoxia and hypoxia were subject to immunoprecipitation with Ubiquitin antibody, followed by immunoblotting. Input is 10% of whole cell lysate. **b** In vivo ubiquitination assay of Akt from 293T cells transfected with WT, K63R and K48R His-Ub in response of 1% $O_2$. Input is 10% of whole cell lysate. **c** Control (shLuc) and AMPKα1 knockdown MDA-MB-231 cells in normoxia and hypoxia were subjected to immunoprecipitation with K63-specific Ubiquitin antibody, followed by immunoblotting. Input is 10% of whole cell lysate. **d–f** In vivo ubiquitination assay of Akt from WT, *AMPKα$^{-/-}$* MEFs restored with vector and WT AMPKα together with transfection of HA-Akt and His-Ub in response of hypoxia (1% $O_2$), glucose deprivation and $H_2O_2$ for indicated time. Input is 10% of whole cell lysate. **g** *WT* and *Skp2$^{-/-}$* MEFs challenged with 1% $O_2$ for 8 h were subjected to immunoprecipitation with K63-specific Ubiquitin antibody, followed by immunoblotting. **h** Immunoblotting of *WT* and *Skp2$^{-/-}$* MEFs that were challenged with 1% $O_2$ for 4 and 8 h. **i** Immunoblotting of control (shLuc) and Skp2 knockdown (#1) MDA-MB-231 cells were serum starved and treated with $H_2O_2$ (100 μM) for indicated time

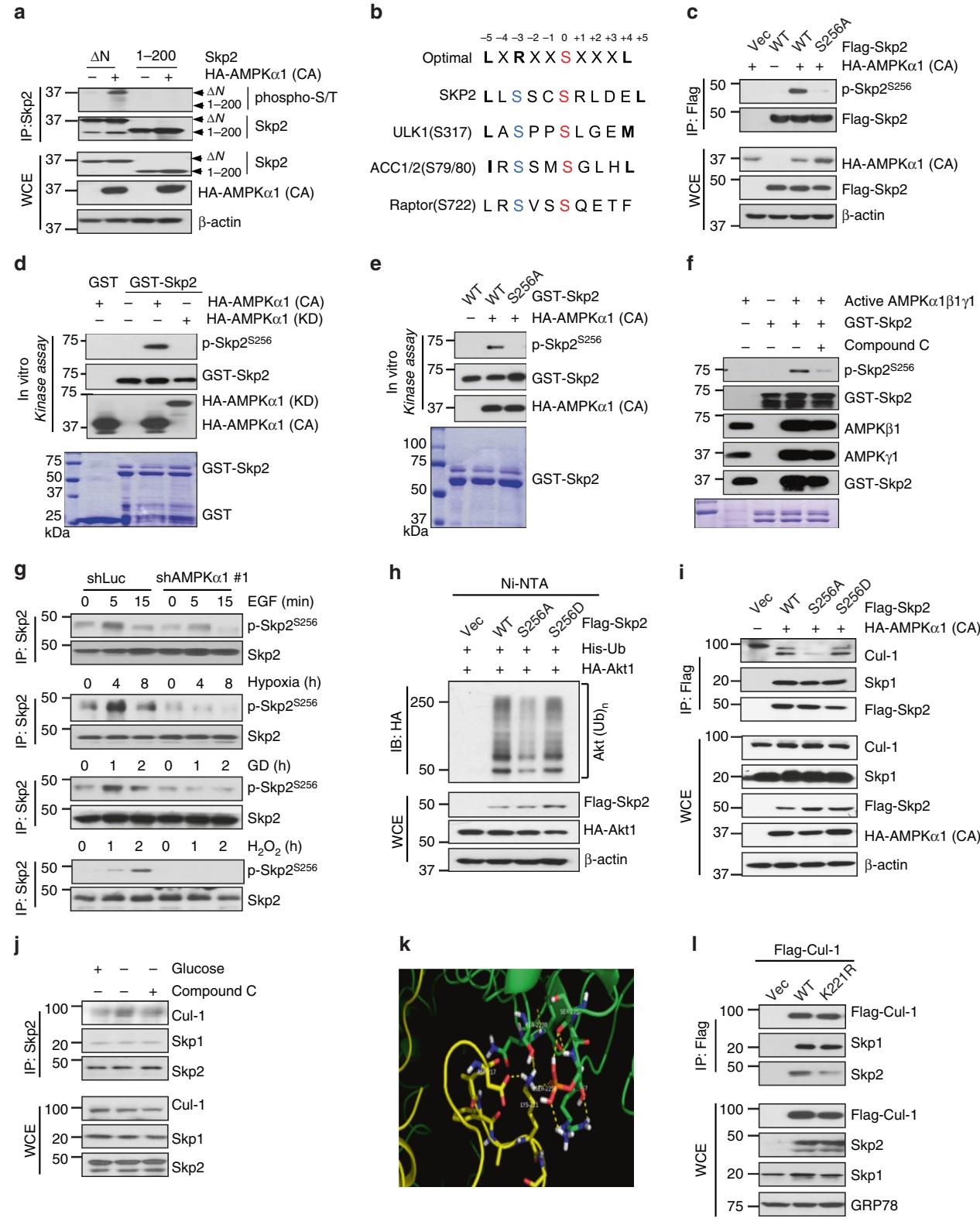

with Cul1, but Skp2 S256A failed to do so (Fig. 4i). To explore Skp2-Cul1 binding under physiological conditions instead of overexpressing active AMPK, we deprived glucose from cells and found that glucose removal promoted the binding of Skp2 with Cul1 and Skp1 in an AMPK dependent manner (Fig. 4j). Our data suggest that AMPK-mediated Skp2 S256 phosphorylation is

critical for the Skp2-Cul1 interaction, thereby promoting Skp2 SCF E3 ligase activity. Indeed, using the computer-guided 3D structure modeling for the Skp2 SCF complex, we found Skp2 S256 residue is directly in contact with Cul1 K221 residue (Fig. 4i). Notably, the interaction of Skp2 with Cul1 K221R was much weaker than the interaction of Skp2 with Cul1 (Fig. 4j).

**Fig. 4** AMPK directly phosphorylates Skp2 at S256 to promote Skp2 SCF complex formation. **a** HEK293 cells transfected with Skp2 ΔN, 1–200 and the indicated HA-AMPKα (CA) constructs were subjected to immunoprecipitation, followed by immunoblotting. **b** The sequence of optimal AMPK motif, Skp2 and known AMPK substrates are listed. The Serine (S) residues that phosphorylated by AMPK are highlighted. **c** HEK293 cells transfected with Skp2 WT, S256, and the indicated HA-AMPKα (CA) constructs were subjected to immunoprecipitation, followed by immunoblotting. **d** Constitutively active (CA) and kinase dead (KD) HA-AMPKα1 proteins purified from 293T cells and recombinant GST-Skp2 WT purified from bacteria were subjected to in vitro kinase assay, followed by Immunoblotting. **e** Constitutively active (CA) HA-AMPKα1 purified from 293T cells and recombinant GST-Skp2 WT, S256A purified from bacteria were subjected to in vitro kinase assay, followed by Immunoblotting. **f** In vitro kinase assay of recombinant GST-Skp2 and active AMPKα1β1γ1 complex purified from sf9 cells with or without Compound C (5 μM) for 30 min followed by Immunoblotting. **g** Immunoblotting of control (shLuc) and AMPKα1 knockdown (#1) MDA-MB-231 cells treated with EGF, hypoxia (1% $O_2$), glucose deprivation and $H_2O_2$ using Skp2 S256 phosphor-specific antibody. **h** In vivo ubiquitination assay in 293T cells transfected with the indicated plasmids was performed, followed by immunoblotting. **i** 293T cells transfected with Flag-Skp2 WT, S256A/D, and the indicated HA-AMPKα (CA) constructs were subjected to immunoprecipitation with Flag antibody, followed by immunoblotting. **j** MDA-MB-231 cells treated with or without glucose free medium and compound C were subjected to immunoprecipitation with Skp2 antibody, followed by immunoblotting. **k** The potential binding pocket on the interface of Skp2-Cullin1. Skp2 is shown in green and Cullin1 is displayed in yellow. **l** 293T cells transfected with Flag-Cullin1 WT and K221R were subjected to immunoprecipitation with Flag antibody, followed by immunoblotting

However, there was no difference in the interaction of Skp1 with WT Cul1 or Cul1 K221R. Thus, it is likely that Skp2-Cul1 interaction can be achieved under certain stress and EGF treated conditions where AMPK is activated to phosphorylate Skp2 at S256, thereby providing the contact sites for Cul1.

**The AMPK-Skp2-Akt axis regulates multiple stress-related and EGF-induced cellular processes, including cell survival, angiogenesis, glycolysis, and migration**. To explore whether our newly identified AMPK-Skp2-Akt axis regulates diverse cellular functions, we examined the role of AMPK and Skp2 in Akt and stress-dependent biological processes. We first tested whether AMPK acts through the Skp2/Akt axis to regulate cancer cell survival under diverse metabolic stresses. As expected, AMPK knockdown reduced cancer cell survival under hypoxia or glucose deprivation, which could be rescued by introduction of Myr-Akt (Fig. 5a, b and Supplementary Fig. 6a). We overexpressed Skp2 WT, S256A or S256D mutant into AMPK knockdown cells and found that expressing Skp2 and Skp2 S256D, but not Skp2 S256A, rescued cancer cell survival under hypoxia or glucose deprivation (Fig. 5c, d and Supplementary Fig. 6b). Thus, AMPK acts through Skp2 S256 phosphorylation and Akt activation to maintain cancer cell survival under hypoxia and glucose deprivation.

Glucose deprivation is shown to induce VEGF production in human monocyte[29]. However, how VEGF production is regulated under glucose deprivation is not well understood. Remarkably, AMPK knockdown or Skp2 knockdown impaired the production of VEGF upon glucose deprivation (Supplementary Fig. 6c, d). Ectopic expression of a constitutively active form of Akt (Myr-Akt) or Skp2 S256D, but not Skp2 S256A, in AMPK knockdown cells rescued the defect in VEGF production upon glucose deprivation (Supplementary Fig. 6e, f). Since VEGF is a growth factor that stimulates vascularization, we examined whether the AMPK-Skp2-Akt axis regulates the ability of cancer cells to induce vascularization by performing an in vitro human umbilical vein endothelial cell (HUVEC) tube formation assay. HUVEC cells cultured with conditioned medium from AMPK knockdown cancer cells displayed lower tube and branch numbers, which could be partially rescued by Skp2 S256D, but not by Skp2 S256A (Fig. 5e and Supplementary Fig. 6g). These results suggest that the AMPK-Skp2-Akt axis regulates VEGF expression, which may in turn drive tumor angiogenesis under glucose deprivation.

EGF regulates a myriad of biological processes, such as glycolysis and cell migration. We then determined whether activation of the AMPK-Skp2-Akt axis contributes to EGF-induced glucose metabolism and cancer cell migration. Notably, AMPK knockdown in cancer cells attenuated EGF-induced glucose uptake and lactate production (Fig. 5f and Supplementary Fig. 6h). Consistently, the induction of the glucose transporter Glut1 expression upon EGF treatment was also impaired in AMPK knockdown cells (Fig. 5g). Restoration of Akt activation by Myr-Akt in AMPK knockdown cancer cells rescued Glut1 expression, glucose uptake and lactate production (Fig. 5f, h and Supplementary Fig. 6i). Moreover, overexpression of Skp2 S256D, but not Skp2 S256A, also rescued Glut1 transcription, glucose uptake, and lactate production in AMPK knockdown cancer cells (Fig. 5i and Supplementary Fig. 6j, k). While EGF-induced cancer cell migration was reduced upon AMPK knockdown, restoration of Myr-Akt or Skp2 S256D, but not Skp2 S256A, in AMPK knockdown cancer cells rescued such defect (Fig. 5j, k and Supplementary 6l). Accordingly, these data suggest that the AMPK-Skp2-Akt axis regulates EGF-induced glucose metabolism and cancer cell migration.

**AMPK-mediated Skp2 S256 phosphorylation promotes cancer progression, predicts poor survival outcome and contributes to the resistance of EGFR targeting therapy**. As AMPK acts through Skp2 S256 phosphorylation and subsequent Akt activation to promote glucose metabolism, cancer cell migration, and VEGF expression in vitro, we then determined whether AMPK displays oncogenic activity in vivo and if so, whether it acts through promotion of Skp2 S256 phosphorylation. Consistently, AMPK knockdown suppressed breast tumor growth in vivo (Fig. 6a and Supplementary Fig. 7a). Remarkably, ectopic expression of Skp2 S256D, but not Skp2 S256A, in AMPK knockdown cells rescued the defect in tumor growth upon AMPK deficiency (Supplementary Fig. 7b). Moreover, overexpression of Skp2 or Skp2 S256D, but not Skp2 S256A, promoted breast cancer development (Fig. 6b and Supplementary Fig. 7c). IHC staining revealed that while S256D mutant was able to induce Akt T308 phosphorylation, Skp2 S256A phosphorylation deficient mutant failed to do so (Supplementary Fig. 7d). We examined whether AMPK-Skp2 axis regulates in vivo vascularization determined by CD34 expression, a vascular marker, in xenograft tumors. CD34 expression was markedly reduced in AMPK knockdown tumor, which could be rescued by Skp2 S256D, but not Skp2 S256A (Supplementary Fig. 7e, f). Hence, our data suggests that AMPK displays oncogenic activity in vivo through promotion of Skp2 S256 phosphorylation.

To determine the clinical importance and relevance of the AMPK-Skp2-Akt axis in human breast cancer, we performed immunohistochemistry (IHC) to detect Skp2 S256 phosphorylation pSkp2 (S256), AMPK T172 phosphorylation (pAMPK) and Akt S473 phosphorylation (pAkt) in 208 patients with various

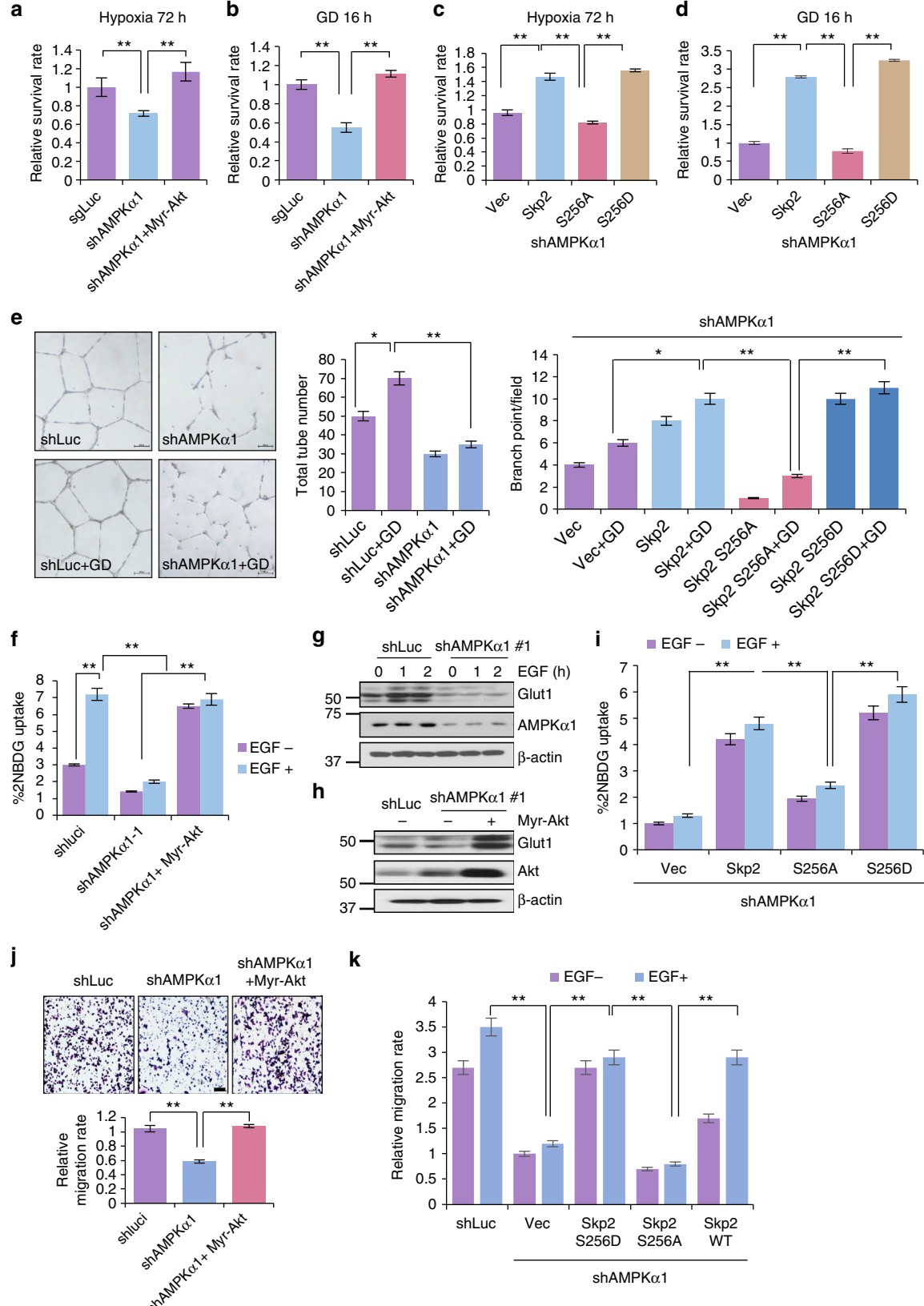

stages of breast cancer. We used the American Joint Committee on Cancer (AJCC) stage to judge the clinical aggressiveness of a tumor at diagnosis. Notably, pAMPK, pSkp2 (S256) and pAkt were highly upregulated in late-stage breast cancer, and the increased expression of both pAMPK and pSkp2 (S256) was significantly associated with pAkt expression in the high stage of breast cancer patients (Fig. 6c–e and Supplementary Table 2). Univariate survival analysis revealed that high expression of

**Fig. 5** AMPK-Skp2-Akt axis is critical for survival under stress and glucose deprivation induced VEGF secretion and EGF-induced glycolysis and migration. **a**, **c** Cell survival analysis of MDA-MB-231 cells with control (shLuc), AMPKα1 knockdown, AMPKα1 knockdown along with Myr-Akt, Skp2 WT, or Skp2 S256A or Skp2 S256D restoration under hypoxia (1% O$_2$) for 72 h. **b**, **d** Cell survival analysis of MDA-MB-231 cells with control (shLuc), AMPKα1 knockdown, AMPKα1 knockdown along with Myr-Akt, Skp2 WT, or Skp2 S256A or Skp2 S256D restoration under glucose deprivation for 16 h. **e** MDA-MB-231 cells with control (shLuc), AMPKα1 knockdown and AMPKα1 knockdown with vector control, Skp2 WT and S256A or Skp2 S256D restoration were treated with a glucose-free medium for 8 h, and supernatant was collected for HUVEC tube formation assay. Each value represents the mean ± SEM in three independent experiments. **P < 0.01. **f** MDA-MB-231 cells with control (shLuc), AMPKα1 knockdown (#1) and Myr-Akt restoration in AMPKα1 knockdown were starved in DMEM glucose-free medium for 4 h and added with 2-NBDG for 30 min. Cells were then subjected to FACS analysis. Each value represents the mean ± SEM (n = 4 per group) in three independent experiments. **P < 0.01. **g** Immunoblotting of control (shLuc) and AMPKα knockdown (#1) MDA-MB-231 cells serum starved and treated with EGF for 15 and 30 min. **h** Immunoblotting for MDA-MB-231 cells with control (shLuc), AMPKα1 knockdown and AMPKα1 knockdown along with Myr-Akt restoration. **i** MDA-MB-231 AMPKα1 knockdown cells with Skp2 WT, S256A or S256D restoration were starved in DMEM glucose-free medium for 4 h and added with 2-NBDG for 30 min. Cells were then subjected to FACS analysis. Each value represents the mean ± SEM (n = 4 per group) in three independent experiments. **P < 0.01. **j** Cell migration assay of MDA-MB-231 cells with control (shLuc), AMPKα1 knockdown and AMPKα1 knockdown along with Myr-Akt restoration. **k** MDA-MB-231 AMPKα1 knockdown cells with Skp2 WT, S256A or S256D restoration were subject to cell migration assay with or without EGF

pAMPK or pSkp2 (S256) significantly reduced breast cancer disease-specific and metastasis-free survival (Fig. 6f, Supplementary Fig. 7g and Supplementary Table 3). We examined endogenous K63-linked ubiquitination of Akt in human breast cancer samples compared with normal counterparts and found Akt K63-linked ubiquitination was much higher in tumor tissues than in adjacent normal tissues and correlated with Akt T308 phosphorylation (Supplementary Fig. 7h), further supporting the role of Akt ubiquitination in Akt activation in breast cancer patients. Thus, our data underscores the important role of the AMPK/Skp2 S256 phosphorylation/Akt axis in human breast cancer progression and disease outcome.

Hyper-activation of PI3K/Akt pathway is associated with acquired drug resistance in many cancer types[30]. Studies using a Gefitinib-sensitive cell line have shown that the resistance to Gefitinib is associated with increased Akt phosphorylation and activation[31]. Given that EGF activates AMPK to drive Skp2-mediated ubiquitination and activation of Akt, we next examined whether the AMPK-Skp2-Akt axis contributes to the resistance to EGFR targeting therapy in non-small cell lung cancer cells (NSCLC). AMPK or Skp2 knockdown in the Gefitinib-resistant H1975 cell line not only impaired EGF-mediated Akt phosphorylation and activation (Supplementary Fig. 8a-d), but also increased the sensitivity of H1975 cells to Gefitinib treatment (Supplementary Fig. 8e). Consistently, pharmacological inactivation of Skp2 by Skp2 inhibitor (compound #25), which disrupts the Skp2 SCF complex and Skp2 E3 ligase activity[32], also enhanced the efficacy of Gefitinib in killing H1975 cells (Supplementary Fig. 8f). Moreover, stable overexpression of Skp2 S256A, but not Skp2 S256D, in H1975 cells heightened the response of H1975 cells to Gefitinib (Fig. 6g and Supplementary Fig. 8g), suggesting that AMPK-mediated Skp2 S256 phosphorylation may cause the resistance of H1975 cells to Gefitinib, and targeting AMPK, Skp2, or AMPK-mediated Skp2 S256 phosphorylation may overcome the resistance of NSCLC cells to Gefitinib.

## Discussion

Stress responses occurring during physiological and pathological conditions such as metabolic stress, hypoxia, and oxidative stress play important dual roles in both cancer prevention and promotion[33–35]. This outcome of tumor phenotypes is achieved through distinct signaling pathways impinging on oncogenic and tumor suppressive activities. Activation of oncogenic Akt induced by these insults likely contributes to cancer cell survival and proliferation ultimately leading to tumor initiation and promotion, but the underlying mechanism accounting for Akt activation

remains largely unclear. We demonstrate in this study that the stress kinase AMPK is a key upstream kinase responsible for oncogenic Akt activation under these diverse stresses. We identify Skp2 as a novel substrate of AMPK whose phosphorylation at S256 induces Skp2 SCF E3 ligase activity to drive K63-linked ubiquitination and activation of Akt in response to diverse stresses. Our study therefore places the AMPK/Skp2 axis as a critical signaling input critical for Akt phosphorylation and activation under diverse stress responses and further underscores the importance of K63-linked ubiquitination of Akt in serving as a common mode for Akt phosphorylation and activation by these stimuli.

The finding that AMPK serves as an upstream kinase necessary for EGF-mediated Skp2 and Akt activation and oncogenic processes is highly unexpected and surprising, since Akt is known to be an upstream kinase important for mTORC1 activation[5]. The balance between AMPK-mediated TSC2 activation/mTORC1 inactivation vs. Akt-mediated mediated TSC2 inactivation/mTORC1 activation determines the final outcome of mTORC1 activation in response to the growth factor EGF. AMPK activation by EGF triggers Skp2 S256 phosphorylation, leading to activation of Skp2 SCF E3 ligase and subsequent Akt ubiquitination and activation. Notably, we show that AMPK activation is required for EGF-mediated biological functions such as glycolysis and cell migration. Nevertheless, AMPK-mediated oncogenic Akt activation provides another molecular explanation of how AMPK could serve as a tumor promoter.

While calcium signaling is elicited by EGF-mediated PLCγ activation[36], its role in EGF-induced Akt activation and biological processes remain elusive. As an important second messenger, intracellular Ca$^{2+}$ level is orchestrated by various calcium pumps, channels and calcium binding proteins and mediates numerous biological functions such as muscle contraction, vascular dilation, neuronal transmitter, fertility, and apoptosis[37]. Thus, calcium homeostasis is tightly controlled to maintain normal cellular functions[38]. However, cancer cells often display altered expression of calcium channels and pumps, resulting in deregulation of calcium signaling that is associated with cancer progression and drug resistance[39]. Surprisingly, we showed that the stress kinase AMPK, normally induced by the stress conditions where ATP levels drop, is also induced by EGF through calcium-dependent activation of CaMKKβ and is essential for EGF-induced Akt K63-linked ubiquitination, phosphorylation and activation. Thus, calcium-dependent CAMKKβ activation is likely an oncogenic signal mediating some of the important functions of EGF. In support of this notion, CAMKKβ knockdown decreases cell proliferation, migration and invasion in prostate cancer cells[40], and CaMKKβ is amplified in prostate cancer, invasive breast

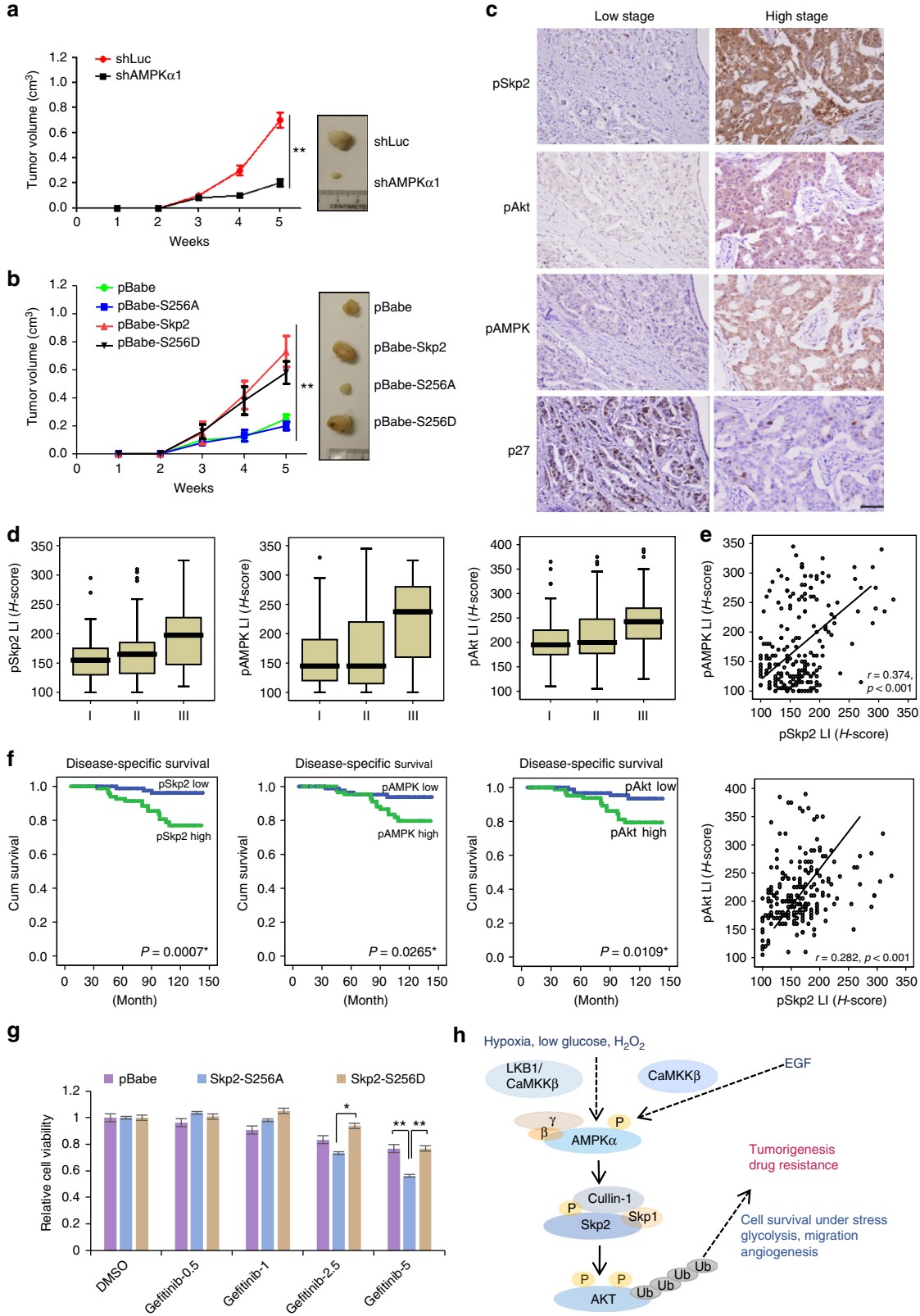

cancer, and ovarian cancer as shown in the TCGA data set[41,42]. Consistently, we found EGF-induced Glut1 expression was impaired in CaMKKβ knockdown cells (Supplementary Fig. 9a), suggesting a role of CaMKKβ in glycolysis under EGF stimulation. While our study places CAMKKβ upstream of AMPK to induce Akt activation under EGF treatment, we cannot rule out

the possibility that CAMKKβ may directly induce Akt activation under EGF treatment, since a recent report showing that CaMKKβ could directly phosphorylate Akt at T308 in a $Ca^{2+}$/CaMKKβ dependent manner in ovarian cancer cells[43].

AMPK is a selective kinase that recognizes a consensus motif in the substrate and prefers leucine and methionine hydrophobic

**Fig. 6** AMPK-mediated Skp2 S256 phosphorylation and Akt activation promotes cancer progression, predicts poor survival outcome and contributes to resistance of EGFR targeting therapy. **a** MDA-MB-231 cells with control (shLuc) and AMPKα1 knockdown were subcutaneously injected into nude mice. Tumor size was measured by the caliper, and the result was showed as means ± SEM. ($n = 5$). **$P < 0.01$. **b** MDA-MB-231 cells with stable expression of the indicated pBabe-Skp2 WT, S256A and S256D were subcutaneously injected into nude mice. Tumor size was measured by the caliper, and the result is shown as means + SEM ($n = 5$). **$P < 0.01$. **c** pSkp2 (S256) and pAMPK are upregulated in breast cancer patients. Representative images of histological analysis of pSkp2 (S256), pAkt, pAMPK and p27 staining in early (Left panel) and late-stage human breast cancer (Right panel). Scale bar, 200 μm. **d** Overexpression of pSkp2 (S256) and pAMPK are detected in advanced stage of breast cancer patient. Box plot represents pSkp2 (S256), pAMPK and pAkt expression in different stages of breast cancer patient. **$P < 0.01$. **e** pAMPK and pAkt expression both correlate with pSkp2 (S256) in breast cancer patients. Scatter plots of pSkp2 (S256) expression vs. pAMPK (Upper panel) and pAkt (Lower panel) expression in breast cancer patient were presented. **f** Overexpression of pAMPK, pSkp2 and pAkt predicts poor survival outcome of breast cancer patients. Kaplan–Meier plots showed that high expression of pAMPK (Left panel), pSkp2 (S256) (Middle panel) and pAkt (Right panel) significantly predicted disease-specific survival. p-values are shown in the graphs. **g** Gefitinib-resistant H1975 NSCLC cells with stable expression of the indicated vector, pBabe-Skp2 S256A and S256D were treated with Gefitinib

residues at −5 and +4 positions and basic residues at −3 and −4 positions of actual phosphorylation site. Acidic residues at +3 and +4 positions are also reported as a secondary motif[44]. We identified Skp2 S256 as a direct AMPK phosphorylation site. Consistent with the optimal AMPK consensus motif, Skp2 also contains a leucine in −5 position, but the other leucine is in +5 instead of +4 position. The serine in −3 position related to S256 shares similarity with other known AMPK substrates such as ACC1/2, Raptor and Ulk[44–46]. Moreover, similar to that in the reported secondary motif which has an acidic residue at +3 position, Skp2 also contains acidic residues at +3(D) and +4(E) positions (Fig. 2b). This comparison indicates the flexibility and complicity of AMPK consensus motif. It may also help predict the potential AMPK substrates using the bioinformatics approach.

The Skp2-SCF complex consists of F-box protein Skp2, Skp1 adapter, Cul1, and Rbx1 E3 ligase. Skp2 contains an N-terminal domain whose function is unclear, an F-box domain known to mediate Skp1 binding and a C-terminal LRR domain responsible for its substrate recognition[47]. The crystal structure study of the Skp2 SCF complex reveals that Skp1 acts as a linker to bridge both Skp2 and Cul1 by binding to the F-box domain of Skp2 in vitro[48]. However, it is unclear whether Skp2 has a direct contact site for Cul1 to form Skp2 SCF complex independently of Skp1 inside cell in vivo, since this crystal study used a truncated F-box domain of Skp2 instead of full-length Skp2[48]. Using computer-based modeling, we uncovered that the Skp2 S256 residue, located inside the LRR domain, could directly contact Cul1 K211 (Fig. 4k). This notion was supported by our finding that Cul1 K211R fails to interact with Skp2, but still retains its ability to interact with Skp1 (Fig. 4l). Moreover, while phosphorylation deficient mutant Skp2 S256A still interacts with Skp1, it fails to interact with Cul1 in vivo (Fig. 4i). Interestingly, phosphorylation-mimetic mutant Skp2 S256D maintains its ability to interact with both Cul1 and Skp1(Fig. 4i). Thus, phosphorylation of Skp2 at S256 by AMPK serves as a critical step important for Cul1 binding, thereby facilitating Skp2 SCF complex and its E3 ligase activity to drive Akt ubiquitination and activation by diverse stresses or growth EGF.

Accumulating evidence reveals that cancer cells cannot grow alone. Without the support from surrounding cells, such as inflammatory cells (e.g. M2 macrophages) and endothelial progenitor cells, cancer cell would not have sufficient growth factors, cytokines and nutrients necessary for their survival, proliferation and migration[49,50]. VEGF is an important tumor-promoting factor that is induced by diverse stresses including hypoxia and metabolic stress and is critical for angiogenesis and subsequent cancer progression and metastasis[51]. However, how metabolic stress induces VEGF remains poorly understood. We present evidence that AMPK-mediated Skp2 S256 phosphorylation and subsequent Akt activation are critical for VEGF induction and secretion in cancer cells upon metabolic stress. Thus, the AMPK-Skp2-Akt axis may serve as an important pathway for cancer cells to modify the tumor microenvironment by promoting VEGF-induced angiogenesis. Remarkably, we found that the AMPK-Skp2-Akt axis is activated in human cancers and that the induction of AMPK-mediated Skp2 S256 phosphorylation significantly predicts poor survival outcome of cancer patients.

In summary, we have uncovered a common mechanism by which diverse stresses as well as the growth facture EGF induce oncogenic Akt ubiquitination, phosphorylation and activation. Surprisingly, AMPK has been identified as an essential factor for Akt ubiquitination, phosphorylation and activation not only upon diverse stresses, but also upon EGF stimulation. Phosphorylation of Skp2 at S256 by AMPK elicits the formation and activation of the Skp2 SCF complex, in turn driving K63-linked ubiquitination and activation of Akt (Fig. 6h). Activation of AMPK/Skp2 S256 phosphorylation/Akt axis in cancer drives cancer progression and drug resistance, leading to poor survival outcome of breast cancer patients. This newly identified AMPK/Skp2/Akt pathway provides not only the molecular basis of how AMPK serves a protective and survival advantage under diverse stresses, but also offers a novel therapeutic strategy for cancer treatment and overcoming acquired resistance to EGFR targeted therapy. This pathway may also extend to other stress conditions, such as ER stress and DNA damage, which trigger oncogenic Akt activation and oncogenic processes via unknown mechanisms.

## Methods

**Cell culture and reagents**. HEK293T, HEK293, MDA-MB-231, HUVEC cells were purchased from American Type Culture Collection. Skp2 WT and KO MEFs were prepared as previously described[6]. WT and AMPKα1/α2 knockout MEFs were from Dr. Kun-Liang Guan from the University of California, San Diego. H1975 Gefitinib resistant NSCLC cell line were a kind gift from Dr. Hui-Wen Chen from National Taiwan University, Taipei, Taiwan. HEK293 and 293T cells were cultured in F12 medium supplemented with 10% FBS. HUVEC cells were maintained in EGM-plus media from Lonza (Cat # CC-5035). All other cells were cultured in DMEM/high glucose medium supplemented with 10% FBS. Kinase inhibitor library (Cat# L1200) and Gefitinib (Cat # S1025) were purchased from Selleck Chemicals. 2 NBDG (Cat # 72987), H₂O₂ (Cat # H3140), compound C (Cat # 171260), STO609 (Cat # S1318) were from Sigma. Plasmids: His6-ubiquitin (His-Ub), pcDNA3-Flag-Skp2, pBabe-Skp2 constructs were previously described[6]. Skp2 and Cul1 mutants were generated using site-direct mutagenesis. The following primers for mutagenesis are used:

Skp2 S256A-F 5′-CTTTGCTAAGCAGCTGTGCCCAGACTGGATGAG,
Skp2 S256A-R 5′-CTCATCCAGTCTGGCACAGCTGCTTAGCAAAG,
Skp2 S256D-F 5′-GCTAAGCAGCTGATAGACTGGATGAG,
Skp2 S256D-R 5′-CTCATCCAGTCTATCACAGCTGCTTAGC,
Skp2 S367A-F 5′-GGATCCGGTTGGACGCTGACATCGGATGCCCTC
Skp2 S367A-R 5′-GAGGGCATCCGATGTCAGCGTCCAACCGGATCC
Skp2 S382A/S383A-F 5′-GAACTTCCAAACTCAAGGCCGCCCATAAGC
TATTTTGC

Skp2 S382A/S383A-F 5′-GCAAAATAGCTTATGGGCGGCCTTGAGTTT
GGAAGTTC
Cul1 K221R-F 5′-GATGATGCATTTGCACGGGGCCCTACGTTAACAG
Cul1 K221R-F 5′-CTGTTAAGGTAGGGCCCCGTGCAAATGCATCATC

**Immunoprecipitation (IP) and immunoblotting (IB).** Cells were lysed in RIPA buffer (50 mM Tris-HCI pH 7.4, 1% NP40, 0.5% Na-deoxycholate, 0.1% SDS, 150 mM NaCI, 2 mM EDTA, and 50 mM NaF). SDS sample buffer was added to the lysate for immunoblotting. For IP, Cells were lysed and quantified for protein concentration. The primary antibody was added and incubated in 4-degree rotation rack overnight. Protein A/G beads were then added for additional 3 h followed by multiple wash step and immunoblotting. The following antibodies were used: anti-phospho-Akt (T308) (Cat # 9275, 1:1000 dilution), anti- phospho-Akt (S473) (Cat # 4060, 1:1000 dilution), anti-phospho-GSK3β (S9) (Cat # 9322, 1:2000 dilution), anti-phospho-PRAS40 (T246) (Cat # 2997, 1:1000 dilution), anti-phospho-S6K (T389) (Cat # 9206, 1:1000 dilution), anti- S6K (Cat # 9202, 1:1000 dilution), anti-phospho-ACC (S79) (Cat # 11818, 1:2000 dilution), anti-phospho-AMPKα (T172) (Cat #50081, 1:1000 dilution), anti-Akt (Cat # 4691, 1:2000 dilution), anti-AMPKα (Cat # 5831, 1:2000 dilution), anti-ACC (Cat # 3676, 1:500 dilution), anti-GSK3β (Cat # 5676, 1:5000 dilution), anti-PRAS40 (Cat # 2691, 1:1000 dilution), anti-K63 Ub (Cat #12930, 1:1000 dilution) and anti-Cullin1(Cat # 4995, 1:1000 dilution) are all purchased from Cell Signaling Technology; anti-Ubiquitin (Cat # sc-8017), anti-CaMKK, 1:1000 dilution β(Cat # SC-100364, 1:1000 dilution), anti-Skp1(Cat # SC-1568, 1:2000 dilution) are from Santa Cruz; anti-HA (Cat # H3663, 1:2000 dilution), anti-Flag (Cat # F3165, 1:2000 dilution), anti-β actin (Cat # A5441, 1:10000 dilution) are from Sigma; anti-Glut1 (Cat # ab652, 1:500 dilution) are from Abcam. The customed phosphor-Skp2 S256 antibody is from Jiaxing Xinda Biotechnology Co.Ltd.

**Viral infection and transfection.** For AMPK, CaMKKβ, LKB1, and Skp2 knockdown, lentiviral packing plasmid and pLKO.1-puro-shRNA constructs (Sigma) were transfected into 293T cells using calcium phosphate method. After 36 h, the supernatant was collected, and MDA231 parental cells were infected and 2 ug/ml puromycin selected for a week. Knockdown efficiency was detected by western blot. For Skp2 WT, S256A, and S256D restore cell, Skp2 was cloned into pLKO.1 vector (Sigma). Mutants were generated by site-direct mutagenesis. The following lentiviral shRNAs (Sigma) are used: shLuc 5′-CGCTGAGTACTTCGA AATGTC; shSkp2-1 5′-GATAGTGTCATGCTAAAGAAT; shSkp2-2 5′-GCCT AAGCTAAATCGAGAGAA; shCaMKKβ 5′-CCGGGTGAAGACCATGATACGT AAACTCGAGTTTACGTATCATGGTCTTCACTTTTT; shLKB1 5′-CATCTACA CTCAGGACTTCAC; shAMPKα1-1 5′-CCGGGTTGCCTACCATCTCATAATAC TCGAGTATTATGAGATGGTAGGCAACTTTTT; shAMPKα1-2 5′-CCGGGT AGCTGTGAAGATACTCAATCTCGAGATTGAGTATCTTCACAGCTACTTT TTTG

For restoration experiments, we transiently transfected 5–10 µg DNA with turbofect (Thermo Fisher, R0531) into $Skp2^{-/-}$ MEF cells, $AMPK\alpha^{-/-}$ MEF cells and CaMKKβ stably knockdown MDA-MB-231 cells, the detailed information was described in each figure legend.

**Recombinant protein purification and in vitro kinase assay.** Recombinant GST-Skp2 (WT, S256A, and S256D) protein was expressed in BL21 bacteria by transformation and induced with IPTG for 18 h at room temperature. The bacteria were lysed by sonication and subjected to protein purification. GST-Skp2 (WT, S256A, and S256D) was purified by using Glutathione-S-agarose beads (Invitrogen), and then eluted with reduced glutathione (Fisher Scientific). 10-kDa-cutoff Centricon (Millipore) were used for concentrating the proteins. Recombinant active AMP-Kα1β1γ1 complex from Sf9 cells were obtained from Abcam (ab79803). HA-AMPK α1 (CA and KD) proteins were purified by transfecting the plasmid into 293T cells, and lysates were collected after 36 h and purified with HA antibody, followed by elution with HA peptide on the protein A/G beads. Purified protein concentration was determined by Bradford assay and aliquoted to Eppendorf tube. Recombinant GST-Skp2, active AMPKα1β1γ1 complex, or HA- AMPKα1 proteins were incubated for 30 min at 30 °C in 20 µl of in vitro kinase reaction buffer (2 mM DTT, 10 mM MgCl₂, 25 mM Tris-HCl at pH 7.5, 0.1 mM Na₃VO₄ and 5 mM β-glycerophosphate, 0.5 mM ATP). The reaction was stopped after indicated time by SDS sample buffer and subjected to immunoblotting analysis.

**In vivo ubiquitination assay.** 293T cells were transfected with HA-Akt, His-Ub and Flag-Skp2 mutant plasmids. Fourty-eight hours later, cells were then lysed using high-stringency denaturing buffer (10 mM imidazole, 0.1 M Na₂HPO₄/NaH₂PO₄ at pH 8.0 and 6 M guanidine-HCl). Ten percent whole cell lysates were used as input control and the other 90% lysates were incubated with nickel-agarose beads for 3 h. Samples were washed carefully for four times and sample buffer is added to elute the protein binding on the nickel beads. Samples were then subjected to immunoblotting analysis with indicated antibodies. 293T cells were transfected with WT, K63R and K48R His-Ub plasmids for 36 h and put into hypoxia chamber (1% O₂) for 4 h. Cells were lysed and in vivo ubiquitination assay was performed as described above.

**Cell survival assay.** 10⁴ of cells were seeded in 12-well plates in triplicates and cultured for 24 h before any treatment. For cell survival under glucose deprivation, the control group was treated with DMEM high glucose medium (10% FBS) and test group was treated with glucose-free medium (10% FBS) for 16 h. Culture medium with the dead cell was collected and attached cells were also trypsinized and combined. Then all cells were centrifuged at 3000 rpm for 10 min. Supernatants were removed and 100 µl PBS were added to each sample. Trypan blue was added to each sample at 1:10 ratio, and cells were then resuspended and 10 µl aliquots were added into hematocounter for cell counting. The ratio of trypan blue stained and unstained cells were counted under a microscope. For cell survival rate under hypoxia, same cells were seeded with the control group in the incubator and test group in a hypoxia chamber (1% O₂) for 72 h. Viable cells under both conditions were counted using the trypan blue exclusion assay under a microscope.

**In vitro cell migration assay.** Trans-well inserts were placed into 24-well plates with 500 µl complete DMEM high glucose medium on the bottom of the well. 10⁵ cells per insert were seeded and cells were incubated for 6–8 h with or without EGF (R&D, Cat# 236-EG-200) 50 ng/ml for migrating along the serum gradient. The medium was then moved and inserts were washed with 1XPBS, fixed with 4% paraformaldehyde on the other side of the bottom for 30 min at room temperature. Q-tips were used to remove any non-migrated cells inside the bottom of the insert and the other side was stained with crystal violet reagent. The stained cells were then counted and quantified.

**Glucose uptake and lactate production assay.** For glucose uptake assay, cells were seeded in 60 mm plates. After 24 h, cells were refreshed with full serum (10% FBS) and glucose-free DMEM (Gibco, Cat# 11966025), while the complete DMEM high glucose medium served as control. After 8 h, cells were treated with 50 µM 2-NBDG for 30 min and glucose uptake rate was quantified by FACS analysis. For lactate production assay, cells were plated in 24-well plate and cultured overnight in low serum medium (0.1% FBS). After serum starvation, cells were treated with or without EGF (100 ng/ml) for overnight. Culture medium was then transferred from each well to Eppendorf tube. Lactate concentration was determined by using lactate test strips and reading under Accutrend Lactate analyzer (Roche). Next, cells were harvested, stained with trypan blue and viable cell numbers were counted directly under the microscope using hemocytometer. The lactate production was determined by lactate concentration/cells and normalized with the rate detected in the control group without EGF.

**Tube formation assay.** HUVEC endothelial cell was used for in vitro tube formation assay. Matrigel with reduced growth factor (Corning, Cat# 354234) was thawed overnight on ice in 4 °C freezer. 250 µl matrigel was then added to 24-well plates and incubated 30 min in 37 °C incubator before seeding cells. 2 × 10⁴ HUVEC cells were seeded each well and cultured with supernatant from indicated cultured cancer cells. After 24 h, tube formation on each well was examined under bright field microscopy.

**In vivo tumorigenesis assay.** 1 × 10⁶ MDA-MB-231 cells with AMPK knockdown with or without Skp2 overexpression (WT, S256A, and S256D) were subcutaneously injected into 4 weeks' female nude mice ($n = 5–6$). The tumor size was measured by the caliper, and the tumor volume was calculated by the equation: $[mm^3] = (length\ [mm]) \times (width\ [mm])^2 \times 1/2$. Xenograft tissues isolated were fixed in 10% formalin overnight and then embedded in paraffin. All manipulations on the animal are under Institutional Animal Care and Use Committee (IACUC) approval protocol.

**Cell viability assay.** Cytotoxicity of Gefitinib was evaluated with cell counting kit 8 (Vita Scientific, Cat# CK04-05). Briefly, H1975 gefitinib-resistant cell was plated into 96-well plate and cultured overnight before drug treatment. Gefitinib with indicated concentrations were treated for 72 h in test group and DMSO was added into control group at the same time, then media was removed and cell kit −8 solution was added to cells. The supernatant was turned yellow after 1–4 h incubation dependent on the cell density. The optimal density was measured at 450 nm on a micro-plate reader. Absorbance value was recorded and normalized to DMSO control group.

**Patients, human materials, and immunohistochemistry.** The human material study is under the approval of the Institutional Review Board of the Chi-Mei Medical Center in Taiwan. Immunohistochemistry was conducted under standard process with primary phospho-Skp2 (S256), phospho-Akt (S473), pAMPK (T172) and p27 antibodies. The secondary antibody was incubated for 30 min and then developed for 5 min with 3-diaminobenzidine. IHC samples were scored manually by Dr. Chien-Feng Li under the multi-headed microscope as previously described[6].

**Statistical analysis.** All data were presented as means +/− SEM. Each experiment was conducted in three independent repeats. The significance is determined by two-tailed and unpaired student t-test. P-values < 0.05 is considered statistically significant.

## Data availability

All relevant data are included in the published article and its supplementary data files.

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

## Acknowledgements

We thank Drs. GK Liang and HW Chen for providing cells. This work was supported by Start-up funds from Wake Forest School of Medicine and NIH RO1 grants (R01 CA182424 and R01CA193813) to H.K.L and Cancer Prevention and Research Institute of Texas grant RP170333 to S.Z. and J.M. We acknowledge the support of the Wake Forest Baptist Comprehensive Cancer Center Cell, Cellular Imaging & flow cytometry Shared of Resource, supported by the National Cancer Institute's Cancer Center Support Grant (P30CA012197). We also acknowledge the supports from Health and welfare surcharge of tobacco products (MOHW107-TDU-B-212-114014, α9-Nicotinic acetylcholine receptor as a molecular target for smoking-induced breast cancers: emphasizing on risk

factor exposure in Taiwan and development coupled with preclinical assessments of antibody based nanodrugs) to C.F. L.

## Author contributions

F.H., D.X., and H.K.L. conceived and designed the experiments; F.H., D.X., Z.C., X.Z., G. J., G.W. W.N.Z., C.X., C.Y.W. performed the experiments. J.M. and S.Z. performed computer modeling of Skp2 SCF complex. C.F.L. performed IHC and survival analysis in breast cancer patient samples; F.H. and H.K.L. organized and analyzed the data and wrote the manuscript.

## Additional information

**Competing interests:** The authors declare no competing interests.

