## [Peer Review File · Nature Communications]

Reviewers' comments:

Reviewer #1 (Remarks to the Author):

The paper entitled "The crucial role of stress kinase AMPK is driving EGF-mediated oncogenic Akt activation, tumorigenesis and anti-EGFR targeting resistance" by Fei Han et. al. reports on the crucial role of AMPK for growth factor (e.g. EGF) - mediated Akt activation and provides convincing mechanistic evidence on the pathway involved (AMPK - Skp2 - Culin - Akt ubiquitination and activation). Results are confirmed in different tumor cell lines, breast cancer samples and Xenograft models under different stimulatory conditions whereby overexpression and knock-down approaches are backed-up by investigations on endogenous proteins. All together an interesting study reporting important observations supported by well-controlled data.

I have only one minor comment to improve the paper:

In the current state, the paper contains a wealth of data, not always easy to follow for the reader. Focusing the paper and streamlining the data, even omitting some of the less important supplementary information would certainly render the paper "more digestible" for the reader.

Reviewer #2 (Remarks to the Author):

Authors of manuscript titled: "The crucial role of stress kinase AMPK in driving EGF-mediated oncogenic Akt activation, tumorigenesis and anti-EGFR targeting resistance" present revised manuscript where they argue that AMPK is essential for EGF-mediated Akt-activation. According to authors, EGF and other stresses lead to release of Ca²⁺ and CaMKK activation, resulting in AMPK phosphorylation. Active AMPK phosphorylates Skp2 E3 ligase resulting in K63-linked Akt ubiquitination and activation.

Overall, manuscript has been greatly improved in both data quality as well as clarity. However, there are several points that authors should address.

1) Given that there are many blots on every Figure, in general for readability it would be helpful to have cell line name under every blot.

2) Most importantly, blots showing phosphorylation of S6 are overexposed (Figure S8B) and this makes it hard to agree with a conclusion that status of pS6 has not been influenced by EGF after AMPK knockdown. The same is true for Figure S8C: pS6 blots are overexposed and it is impossible to make any conclusion about status of S6 phosphorylation. Therefore, conclusion that "AMPK does not impact mTORC1 activity in response to stresses and EGF" is not supported by the data presented.

3) Do all stresses that lead to AMPK activation in various cell lines have an influence on AMP/ATP ratio in the cells? Or is entire effect on AMPK phosphorylation due to increase in Calcium flux and CaMKK phosphorylation?

4) While Compound C is a great tool for examining whether loss of AMPK activity is involved in the biological processes, it is not very specific (PMID 17850214) and can have AMPK-independent effects on cells (PMID 24419061). Did authors test if Compound C can reduce signaling from various stresses in shAMPK cells?

5) Figure 5A and 5B show that AMPK is required for cell survival during hypoxia and glucose deprivation. I don't think this data really helps too much here and furthermore, this has been published (PMID 15640157, PMID 21570709).

For Figure 5C, 5D, 5I it would be helpful to have shLuc on the same graph (similar to Figure 5K).

6) Supplementary Fig 6G doesn't necessary show much higher levels of Akt K63-Ub in tumors compared to normal cells, although there certainly seems to be a trend. In addition, it is not clear where is correlation to Akt T308 phosphorylation shown?

Reviewer #4 (Remarks to the Author):

SUMMARY: The authors propose an AMPK-dependent signaling pathway through which AKT can be activated by cellular stresses including hypoxia, glucose starvation, and ROS and also by the growth factor EGF. Specifically, this pathway consists of AMPK phosphorylating and activating the E3 Ubiquitin Ligase SKP2 which then ubiquitinates AKT to promote its activation (ubiquitination is nonproteolytic). Activation of AKT via this pathway is proposed to support cancer cell survival and tumor progression. This study builds off of the author's previous study (Chan et al., Cell 2012) which showed that SKP2 could ubiquitinate Akt promoting its activation and that this was required for full activation of Akt by EGF. Therefore the novel points about this Akt activation mechanism in this manuscript include that AMPK phosphorylates and activates SKP2 and that this mediates AKT activation by EGF and the cellular stresses listed above.

This study contains some promising data but currently falls short of connecting all the dots to support their proposed mechanism. For instance, there is no data showing that the S256A-SKP2 mutant affects Akt activation/signaling and there is no data showing that an AKT mutant that can't be ubiquitinated by SKP2 is resistant to the effects of AMPK. Some additional controls and complementary experiments are needed to strengthen the mechanistic figures. Detailed comments are listed below:

GENERAL COMMENTS:

1) AMPK-inhibition/deficiency is likely to lead to increased activation of mTORC1 (due to relief of AMPK-dependent TSC2 and Raptor inhibitory phosphorylation). Increased mTORC1 activation could in turn suppress Akt activation through the multiple mTORC1-dependent negative feedback loops that inhibit PI3K-mTORC2-Akt signaling. While the authors address this issue to some extent in the text and Supp Fig 8 they need to do more to rule this pathway out. The rapamycin treatments show that mTORC1 signaling can affect Akt activation in these cells. Monitoring S6 phosphorylation is generally not sufficient to determine effects on mTORC1 – while the phospho-S6 antibody used here is not listed in the methods, the most common site monitored, S235/236 can be phosphorylated by RSK-downstream of ERK as well as S6K downstream of mTORC1. A better and direct readout of mTORC1 signaling is phospho-T389-S6K. The effects of shAMPK knockdown and treatments on phospho-T389-S6K and total S6K should be shown in Sup Fig 8 instead of S6. It could also be shown whether there are any changes in the mTORC1 feedback substrate Grb10-S476 (antibody from cell signaling) which would affect RTK-Akt signaling. One problem is that since mTORC1 activity is not fully sensitive to rapamycin and since mTORC1 directly phosphorylates Grb10, rapamycin may not fully rescue phospho-Akt in AMPK-deficient cells. Admittedly, this can be a difficult pathway to rule out since there is no specific catalytic inhibitor of mTORC1 that would not also hit mTORC2. But the authors need to more clearly rule out this pathway for readers.

2) Compound C should not be used for the more detailed mechanistic experiments. The literature indicates that Compound C is far from being a specific inhibitor of AMPK. Some publications even suggest Compound C can inhibit mTOR-Akt signaling in an AMPK-independent manner. A kinase inhibitor database (<http://www.kinase-screen.mrc.ac.uk/kinase-inhibitors>) indicates that a couple dozen kinases are inhibited by Compound C to a similar or greater extent than AMPK including kinases that could affect Akt signaling. The use of Compound C should be reserved for the initial screen and making the most basic observations (Figures 1 and 2). It seems quite useless for later

mechanistic experiments where the authors are attempting to establish stronger conclusions about the specific role of AMPK (Fig 3, Sup Fig 3, and Fig 4) The Compound C data from these later figures should be removed because at best its weak support for the thesis and at worst potentially misleading. (with the exception of Fig 4f in which its use is ok since it's an in vitro assay).

3) More experiments should make use of AMPK addback controls in AMPK-deficient cells. Only one experiment makes use of the AMPK α -null MEFs with addbacks (Fig 1d) – why aren't these used in other experiments. The shAMPK knockdown cells are more commonly used but there is no addback control for these. Either use the AMPK α -null MEF addbacks for more key experiments or make the addbacks for the shAMPK knockdown cells.

4) Readout of cellular Akt signaling should include one or two consistent substrates across all figures for which the point is to evaluate the effects on Akt signaling. Phospho-T308 and -S473 are shown in every figure of this type which is good. But not all figures include an Akt substrate to help interpret the effects on Akt signaling. In one figure both GSK3 and PRAS40 are shown, in some figure only GSK3 is shown, and in other figures no substrate is shown. Ideally, both PRAS40 and GSK3 phosphorylation status would be consistently shown in every figure in which the effects of a treatment on Akt activity/signaling must be interpreted. If the authors only want to show one substrate, PRAS40 may be best since it is a more specific target of Akt compared to GSK3 (which can be phosphorylated on the same sites by the related kinases S6K and RSK that might also be affected by the treatments used here). The same readouts for Akt should be shown in every figure.

5) Fix misaligned labeling of figures. For example in 1D, 2A, 4A, 4D, 4E, Sup4A, etc.

FIGURE-SPECIFIC COMMENTS:

Figure 1) Basic observation about requirement of AMPK for hypoxia, glucose starvation, ROS-induced Akt is convincing but the evidence should be strengthened. The evidence for the basic observation that stress-induced Akt activation is diminished in AMPK-deficient cells is fairly convincing since complementary data from an inhibitor, shAMPK-knockdown cells, and AMPK α null MEFs is shown. 1D is a good experiment with the proper addback controls. However, Figure 1 as a whole should be strengthened by including the addback controls (WT and KD AMPK addback) used in 1D as controls in 1E (glucose starvation) and 1F (ROS) since these are distinct stimuli being tested for the first time. Ideally, addback rescue controls would have been included for the shAMPK α 1 cells also. Compound C should be used with caution as it is not a very specific inhibitor but it is fine here when used simply to establish the basic observation that AMPK appears to be required for full Akt activation in these settings. Compound C treatments after glucose deprivation and H₂O₂ treatment should also be shown in Figure 1 for uniformity.

Figure 2) The data supporting the basic observation about requirement of AMPK for EGF signaling should be strengthened.

2-1. This figure should mirror the evidence in Fig 1 a bit more (not including the original kinase inhibitor screen). The cells from Fig 1D (WT, AMPK α -/-+Vec, +WT, +KD) should be used to test the requirement of AMPK for EGF-stimulated Akt. Its good to test the effects in other cell lines but if room is tight, 2B could go in supp data as this type of data did in Fig1. The authors should also test the effects of Compound C on EGF treatments. Again, Compound C is not very specific but is ok to use here because a basic observation is still being established (that AMPK influences EGF signaling to Akt).

2-2. The phosphorylation of the EGFR and ERK should be shown throughout this figure to establish whether AMPK loss is having upstream or downstream effects on EGF signaling other than on Akt. This will help readers evaluate the validity of the proposed mechanism of AMPK-dependent effects

on EGF-stimulated AKT activation.

2-3. It's nice that the CAMKKb data consists of complimentary shRNA-knockdown and inhibitor experiments. However, for the shRNA knockdown experiment the shLUC, shCAMKKb, and shLKB1 should be done side by side in the same experiment and run on the same gel to make a single subfigure (instead of in separate figures 2c and sup 2C) Only in this way can the reader properly compare the effects of the two upstream kinases on AMPK signaling. Following this experiment, ideally, a second separate a CAMKKb addback rescue experiment should be included to strengthen the data on involvement of CAMKKb since this point is specifically included in the abstract.

Figure 3) This figure should include data on glucose deprivation and oxidative stress, should eliminate use of Compound C, and should make use of addback control cells.

3-1. The authors specifically mention glucose deprivation and oxidative stress (in addition to hypoxia) in the abstract. Figures showing the effects on Akt ubiquitination in response to glucose deprivation and H₂O₂ should therefore be in the main figure. These should be done with knockdown/knockout cells with addback controls. Compound C should not be used in these experiments to implicate AMPK because it is too non-specific and we have moved beyond basic observation experiments.

3-2. Add an experiment with the AMPK-null MEFs (including addback controls as in 1d to strengthen this figure.

3-3. Not clear why an Akt IP wasn't done instead of Ub IP (as they did in their 2012 paper)? At least in that case it could be shown that total Akt levels aren't changing in the IP but Ub levels are.

3-4. Fig 3b is nice but how do we know that WT, K63R, and K48R Ub are expressed at similar levels in the cell? Differences in expression levels could also explain these observations. Transfected Ub levels need to somehow be shown.

3-5. AKT mislabeled? –In 3D and 3E is total Akt really from the IP? Seems to be mislabeled and is really from WCE.

3-6. In 3f, total Akt is overexposed. Akt is the main thing we're interested in. Why is it overexposed?? Need to see whether Akt levels are even across cell lines.

Figure 4) 4f is a good experiment and the sequence of SKP2-S256, while differing from the main AMPK consensus motif, looks similar to some other AMPK substrates. But the constitutively active AMPK is a problematic reagent and experiments with addback controls are needed.

4-1. Add an experiment with the AMPK-null MEFs (including addback controls as in 1d) to back up shAMPK data and strengthen this figure.

4-2. Put the SKP2 sequence alignment figure from Rebuttal, Response #2 into main figure 4! Show alignment of SKP2-S256 and surrounding sequence with the consensus and secondary consensus AMPK motifs from Gwinn et al Mol Cell 2008 along the top (including position numbering). Importantly, add the well-established substrates Raptor S722 and ACC1/2 S79/80 which both also have an "S" at the -3 position instead of an R! Put these, along with ULK1 right under SKP2 sequence in the alignment. Get rid of NO synthase from this figure – doesn't really help at all. Add references to this subfigure for each substrate. Directly address the differences in the SKP2 sequence compared to the optimal motif pointing out that an "S" at the -3 position is tolerated in other accepted substrates. Also explicitly point out the L is in the +5 position not the +4 position.

Also, there is an acidic residue preceding the L at +3(D) and +4(E) position which is consistent with the +3D/E in the secondary motif listed in Gwinn et al. Readers in the signaling field will appreciate this and will be more likely to accept this as an AMPK substrate.

4-3. Results with Constitutively Active mutant of AMPK are questionable. The constitutively active mutant used here is not sufficiently described in the methods but seems to be a truncated version of AMPK lacking its autoinhibitory domain. There are two problems here. A) Not only is this mutant lacking a big chunk of the endogenous alpha subunit, this mutant does not associate with the beta and gamma regulatory subunits of the full heterotrimeric AMPK complex. The other subunits are required for proper localization and also may affect interactions with substrates. In cells, the CA mutant will be diffusely mislocalized and even in in vitro assays the lack of beta and gamma subunits may allow the alpha subunit to phosphorylate sites it might not otherwise target in the WT complex. Either way, the truncated CA AMPK has the potential to yield misleading results. B) When used in the same experiment, the levels of CA and KD AMPK, which are very different sizes, are shown on separate panels making it impossible to judge their relative expression levels. CA and KD AMPK need to be at similar molar amounts for KD to serve as a proper control. Other papers have used WT AMPK successfully in cells and in vitro kinase assays. Why not use WT AMPK?

4-4. An important point to be drawn from 4G is how the timing of SKP2-S256 phosphorylation correlates with the timing of AMPK-T172 phosphorylation (and ACC phosphorylation) plus AKT phosphorylation in response to these stimuli. Phospho-T172-AMPK, (maybe phospho-ACC), and Phospho-T308 and -S473-Akt should be shown in the WCE lysate for all of these so that we can see them in the same experiment and evaluate whether their timing makes sense.

4-5. The most current version of Scansite, Scansite4 (<https://scansite4.mit.edu/4.0/>) does not predict the same putative AMPK sites on SKP2 as listed here (other than S256). Also, it does not predict any AMPK sites on SKP2 until it is placed on the two lowest stringency settings (Low and Minimum). These differences will be confusing to readers and should be rectified some how. Also, it should be stated that scansite was used on the lowest stringency settings (which is fine given the sequence alignments with Raptor and ACC).

4-6. There are no data showing that S256A or D mutants affect Akt phosphorylation/signaling (e.g. S256A or D addbacks to SKP2-null MEFs and show phospho-Akt, phospho-PRAS40, phospho-GSK3) This is critical to complete the mechanistic link from AMPK to Akt that is the focus of this study as described in the title and abstract. (Note: an experiment with an Akt mutant that could not be Ubiquitinated by SKP2, and hence lost sensitivity to AMPK, would be the final piece of data needed to complete the pathway)

Figures 5 and 6) While there is compelling in vivo data with the SKP2-S256A and D mutants. There is also a lack of data connecting the in vivo effects of SKP2 mutants to AKT in either Fig 5 or Fig 6.

MINOR COMMENTS

- 1) Line up labels better on some figures! Recheck labeling on all figures.
- 2) [page 6] maybe "the kinase inhibitor library" should be "a kinase inhibitor library"?
- 3) [page 6] Rephrase "Activation of PI3K, PDK1, PIM, and Src... were all required for Akt activation..." to say something like "PI3K, PDK1, PIM and Src inhibitors all blocked Akt activation" since some of these inhibitors are not extremely specific and alone can not be used to conclude those targets are the key (just being conservative here – I know there's literature on each of these inhibitors/targets).
- 4) [Fig 1D]. should show total endogenous AMPK levels to demonstrate KO (not just HA-tagged AMPK levels)
- 5) [Fig 2e] Legend for 2e should say how long EGF treatment was.
- 6) FIG 2] Mislabeling in legend. (d) is listed twice meaning last two subfigure legends are

mislabeled.

7) There does appear to be some effect on Akt activation in shAMPKα cells in response to IGF (its just that the figure seems to be overexposed). There even seems to be a bit of an effect on PDGF-stimulated AKT. Perhaps soften the language that AMPK is “dispensable for Akt phosphorylation and activation by IGF-1 and PDGF...”. This doesn’t seem to be completely true.

8) CAMKKβ has been reported to directly phosphorylate T308-Akt directly. While the evidence for this is quite thin, the authors might want to very briefly address this in the discussion.

Response to reviewer's comments

Reviewer #1 (Remarks to the Author):

The paper entitled “The crucial role of stress kinase AMPK is driving EGF-mediated oncogenic Akt activation, tumorigenesis and anti-EGFR targeting resistance” by Fei Han et. al. reports on the crucial role of AMPK for growth factor (e.g. EGF) - mediated Akt activation and provides convincing mechanistic evidence on the pathway involved (AMPK-Skp2-Culin-Akt ubiquitination and activation). Results are confirmed in different tumor cell lines, breast cancer samples and Xenograft models under different stimulatory conditions whereby overexpression and knock-down approaches are backed-up by investigations on endogenous proteins. All together an interesting study reporting important observations supported by well-controlled data.

I have only one minor comment to improve the paper:

In the current state, the paper contains a wealth of data, not always easy to follow for the reader. Focusing the paper and streamlining the data, even omitting some of the less important supplementary information would certainly render the paper “more digestible” for the reader.

We appreciate the reviewer for recognizing the importance of our discovery. We also thank reviewer for the valuable advice. As suggested, we have rearranged our dataset by providing more direct evidence in the main figure and deleted less relevant supplementary data, for example, we have removed the data using compound C as an AMPK inhibitor for Akt ubiquitination study, since compound C has many targets other than AMPK.

Reviewer #2 (Remarks to the Author):

Authors of manuscript titled: “The crucial role of stress kinase AMPK in driving EGF-mediated oncogenic Akt activation, tumorigenesis and anti-EGFR targeting resistance” present revised manuscript where they argue that AMPK is essential for EGF-mediated Akt-activation. According to authors, EGF and other stressed lead to release of Ca²⁺ and CaMKK activation, resulting in AMPK phosphorylation. Active AMPK phosphorylates Skp2 E3 ligase resulting in K63-linked Akt ubiquitination and activation.

Overall, manuscript has been greatly improved in both data quality as well as clarity. However, there are several points that authors should address.

1) Given that there are many blots on every Figure, in general for readability it would be helpful to have cell line name under every blot.

Thanks for reviewer's suggestion. In the revised manuscript, we have marked every blot with cell line name to avoid confusion except for overexpressing experiments done in HEK293T cells in Figure 4.

2) Most importantly, blots showing phosphorylation of S6 are overexposed (Figure S8B) and this makes it hard to agree with a conclusion that status of pS6 has not been influenced by EGF after AMPK knockdown. The same is true for Figure S8C: pS6 blots are overexposed and it is impossible to make any conclusion about status of S6 phosphorylation. Therefore, conclusion that “AMPK does not impact mTORC1 activity in response to stresses and EGF” is not supported by the data presented.

We apologized for the unconvincing data due to overexposure. To further validate the notion that AMPK does not impact on mTORC1 signaling in response of EGF and stresses, we examined another two well-known mTORC1 downstream effectors: phospho-S6K and phospho-Grb10. Consistently, we found both of them were similarly induced upon EGF, glucose deprivation and H₂O₂ treatment in control and AMPK deficient MDA231 cells (Revised Supplementary Figure 8b-d).

3) Do all stresses that lead to AMPK activation in various cell lines have an influence on AMP/ATP ratio in the cells? Or is entire effect on AMPK phosphorylation due to increase in Calcium flux and CaMKK phosphorylation?

The change in AMP/ATP ratio is one of many reasons that triggers AMPK phosphorylation and activation. Reports show that AMPK activation is also detected without change in AMP/ATP ratio upon stresses. For instance, it is shown that the increase of ROS level is sufficient to induce AMPK activity without change in AMP/ATP ratio¹. However, we would not claim that AMPK activation induced by various stresses is entirely dependent on calcium flux and CaMKK β phosphorylation. It remains to be determined which mechanism plays a major role or whether there are other mechanisms involved to trigger AMPK activation.

4) While Compound C is a great tool for examining whether loss of AMPK activity is involved in the biological processes, it is not very specific (PMID 17850214) and can have AMPK-independent effects on cells (PMID 24419061). Did authors test if Compound C can reduce signaling from various stresses in shAMPK cells?

We agree with the reviewer that compound C has also other targets besides AMPK. We therefore used AMPK knockdown and knockout cells to verify our findings. To answer reviewer’s question, we tested the effect of compound C on Akt signaling in WT and AMPK null MEFs under stress conditions and found that while AMPK deficiency inhibited Akt activation under hypoxia, compound C could not further inhibit phosphorylation level of Akt and its downstream GSK3 β and PRAS40 (Revised Fig 1).

5) Figure 5A and 5B show that AMPK is required for cell survival during hypoxia and glucose deprivation. I don't think this data really helps too much here and furthermore, this has been published (PMID 15640157, PMID 21570709). For Figure 5C, 5D, 5I it would be helpful to have shLuc on the same graph (similar to Figure 5K).

We agree with the reviewer that the function role of AMPK in regulating survival under hypoxia and glucose deprivation has been established. However, it remains to be determined whether such effect acts through Akt activation. Figure 5A and 5B showed that restoration of active Akt rescued survival, indicating the importance of AMPK-Skp2-Akt signaling in stress related survival. We separated Figure 5A/C, 5B/D and 5F/I for better presentation. We agree with the reviewer that showing shLuc on the same graph may be more informative, but we think the data should be sufficient to conclude that Skp2 S256 phosphorylation regulates cell survival since we have presented the relevant control in the experiments and vector, WT Skp2, Skp2 S256A and Skp2 S256D were included for the comparison.

6) Supplementary Fig 6G doesn't necessary show much higher levels of Akt K63-Ub in tumors compered to normal cells, although there certainly seems to be a trend. In addition, it is not clear where is correlation to Akt T308 phosphorylation shown?

As suggested, we examined Akt T308 phosphorylation in the same patient samples and found that they were correlated well with K63-Ub level in all 12 breast cancer patients (Revised Supplementary Fig. 6g).

Reviewer #4 (Remarks to the Author):

SUMMARY: The authors propose an AMPK-dependent signaling pathway through which AKT can be activated by cellular stresses including hypoxia, glucose starvation, and ROS and also by the growth factor EGF. Specifically, this pathway consists of AMPK phosphorylating and activating the E3 Ubiquitin Ligase SKP2 which then ubiquitinates AKT to promote its activation (ubiquitination is nonproteolytic). Activation of AKT via this pathway is proposed to support cancer cell survival and tumor progression. This study builds off of the author's previous study (Chan et al., Cell 2012) which showed that SKP2 could ubiquitinate Akt promoting its activation and that this was required for full activation of Akt by EGF. Therefore the novel points about this Akt activation mechanism in this manuscript include that AMPK phosphorylates and activates SKP2 and that this mediates AKT activation by EGF and the cellular stresses listed above. This study contains some promising data but currently falls short of connecting all the dots to support their proposed mechanism. For instance, there is no data showing that the S256A-SKP2 mutant affects Akt activation/signaling and there is no data showing that an AKT mutant that can't be ubiquitination by SKP2 is resistant to the effects of AMPK. Some additional controls and complementary experiments are needed to strengthen the mechanistic figures. Detailed comments are listed below:

Thanks for recognizing the novelty of our study and for offering constructive comments that help further strengthen our study. We have taken the reviewer's comments by heart and addressed all these concerns with relevant experiments (see below).

GENERAL COMMENTS:

1) AMPK-inhibition/deficiency is likely to lead to increased activation of mTORC1 (due to relief of AMPK-dependent TSC2 and Raptor inhibitory phosphorylation). Increased mTORC1 activation could in turn suppress Akt activation through the multiple mTORC1-dependent negative feedback loops that inhibit PI3K-mTORC2-Akt signaling. While the authors address this issue to some extent in the text and Supp Fig 8 they need to do more to rule this pathway out. The rapamycin treatments show that mTORC1 signaling can affect Akt activation in these cells. Monitoring S6 phosphorylation is generally not sufficient to determine effects on mTORC1 – while the phospho-S6 antibody used here is not listed in the methods, the most common site monitored, S235/236 can be phosphorylated by RSK-downstream of ERK as well as S6K downstream of mTORC1. A better and direct readout of mTORC1 signaling is phospho-T389-S6K. The effects of shAMPK knockdown and treatments on phospho-T389-S6K and total S6K should be shown in Sup Fig 8 instead of S6. It could also be shown whether there are any changes in the mTORC1 feedback substrate Grb10-S476 (antibody from cell signaling) which would affect RTK-Akt signaling. One problem is that since mTORC1 activity is not fully sensitive to rapamycin and since mTORC1 directly phosphorylates Grb10, rapamycin may not fully rescue phospho-Akt in AMPK-deficient cells. Admittedly, this can be a difficult pathway to rule out since there is no specific catalytic inhibitor of mTORC1 that would not also hit mTORC2. But the authors need to more clearly rule out this pathway for readers.

We agree with reviewer that we need more readouts to rule out mTORC1 pathway in order to claim the importance of AMPK-Skp2 axis in Akt activation. We have added the information for S6 phosphorylation site (S235/236) in the method section (IP and IB). In supplemental Fig. 8a, rapamycin seemed to inhibit hypoxia-induced phospho-S6 very efficiently. Therefore, we believe that mTORC1 is sensitive to rapamycin, which further triggers positive feedback loop to induce Akt activation. To better monitor mTORC1 activity, we used both phospho-S6K(T389) and phospho-Grb10(S476) as readouts. Results show that mTORC1 activation was not enhanced in AMPK deficient MDA231 cells under EGF, glucose deprivation and H₂O₂ treatment (Revised Supplementary Fig. 8b-c). However, we detected impaired Akt activation in the same cell line under the same condition. Therefore, we rule out the possibility that AMPK mediates Akt activation through upregulating mTORC1.

2) Compound C should not be used for the more detailed mechanistic experiments. The literature indicates that Compound C is far from being a specific inhibitor of AMPK. Some publications even suggest Compound C can inhibit mTOR-Akt signaling in an AMPK-independent manner. A kinase inhibitor database (<http://www.kinase-screen.mrc.ac.uk/kinase-inhibitors>)<https://urldefense.proofpoint.com/v2/url?u=http-3A_www.kinase-2Dscreen.mrc.ac.uk_kinase-2Dinhibitors&d=DwMGAg&c=1jSOPko_E6DLthr5DiSb_42CaaM9-

[nu9QAD4dem7yWs&r=cotLUCHF_IQYKz7sBxqp6sDh4UjpetQaj4cPFbRp6EQ&m=LMg1s3cU5a5B6o8LE0CbI9jnzKME1C56H2xmjrlgYW0&s=cqgoOfjAo6GSQk78tSzkOdOstU5C6AOpz3v6HO8qXLA&e=>](https://pubmed.ncbi.nlm.nih.gov/31111111/)) indicates that a couple dozen kinases are inhibited by Compound C to a similar or greater extent than AMPK including kinases that could affect Akt signaling. The use of Compound C should be reserved for the initial screen and making the most basic observations (Figures 1 and 2). It seems quite useless for later mechanistic experiments where the authors are attempting to establish stronger conclusions about the specific role of AMPK (Fig 3, Sup Fig 3, and Fig 4) The Compound C data from these later figures should be removed because at best its weak support for the thesis and at worst potentially misleading. (with the exception of Fig 4f in which its use is ok since it's an in vitro assay).

We agree with reviewer that Compound C is not entirely specific for AMPK targeting. We have removed compound C data for mechanistic study, as suggested.

3) More experiments should make use of AMPK addback controls in AMPK-deficient cells. Only one experiment makes use of the AMPK α -null MEFs with addbacks (Fig 1d) – why aren't these used in other experiments. The shAMPK knockdown cells are more commonly used but there is no addback control for these. Either use the AMPK α -null MEF addbacks for more key experiments or make the addbacks for the shAMPK knockdown cells.

As suggested, we have restored WT and kinase-dead AMPK in AMPK-null MEFs and examined Akt activation upon various stress conditions (Glucose deprivation and H₂O₂). We observed that restoration of WT AMPK, but not kinase-dead AMPK, rescued Akt activation under glucose deprivation and H₂O₂ treatment (Revised Fig. 1d, e).

4) Readout of cellular Akt signaling should include one or two consistent substrates across all figures for which the point is to evaluate the effects on Akt signaling. Phospho-T308 and -S473 are shown in every figure of this type which is good. But not all figures include an Akt substrate to help interpret the effects on Akt signaling. In one figure both GSK3 and PRAS40 are shown, in some figure only GSK3 is shown, and in other figures no substrate is shown. Ideally, both PRAS40 and GSK3 phosphorylation status would be consistently shown in every figure in which the effects of a treatment on Akt activity/signaling must be interpreted. If the authors only want to show one substrate, PRAS40 may be best since it is a more specific target of Akt compared to GSK3 (which can be phosphorylated on the same sites by the related kinases S6K and RSK that might also be affected by the treatments used here). The same readouts for Akt should be shown in every figure.

We agree with the reviewer that two readouts are indispensable in monitoring Akt activation under particular conditions. We have repeated the experiments and included both phospho-GSK3 β and phospho-PRAS40 reflecting Akt activation along with phospho-Akt T308 and S473 under all conditions-hypoxia, glucose deprivation, H₂O₂ and EGF in the results, and the similar results were obtained by examining phospho-GSK3 β or phospho-PRAS40 signal. At least one consistent readout was shown thereafter.

5) Fix misaligned labeling of figures. For example, in 1D, 2A, 4A, 4D, 4E, Sup4A, etc.

We thank the reviewer for pointing our mistakes. We have corrected these errors in revised Figures.

FIGURE-SPECIFIC COMMENTS:

Figure 1) Basic observation about requirement of AMPK for hypoxia, glucose starvation, ROS-induced Akt is convincing but the evidence should be strengthened. The evidence for the basic observation that stress-induced Akt activation is diminished in AMPK-deficient cells is fairly convincing since complementary data from an inhibitor, shAMPK-knockdown cells, and AMPKalpha null MEFs is shown. 1D is a good experiment with the proper addback controls. However, Figure 1 as a whole should be strengthened by including the addback controls (WT and KD AMPK addback) used in 1D as controls in 1E (glucose starvation) and 1F (ROS) since these are distinct stimuli being tested for the first time. Ideally, addback rescue controls would have been included for the shAMPKa1 cells also. Compound C should be used with caution as it is not a very specific inhibitor but it is fine here when used simply to establish the basic observation that AMPK appears to be required for full Akt activation in these settings. Compound C treatments after glucose deprivation and H₂O₂ treatment should also be shown in Figure 1 for uniformity.

We thank the reviewer for the thoughtful suggestions. As suggested, we have restored WT and kinase-dead AMPK into AMPK-null MEFs and examined Akt activation under glucose deprivation and H₂O₂. We found that phospho-Akt T308 and S473 were rescued by WT but not kinase-dead AMPK addbacks upon glucose deprivation and H₂O₂ treatments (Revised Fig. 1d, e). We also treated MDA-MB-231 cells with Compound C under glucose deprivation and H₂O₂ and observed impaired Akt phosphorylation under these conditions (Revised Supplementary Fig. 1i, j).

Figure 2) The data supporting the basic observation about requirement of AMPK for EGF signaling should be strengthened. This figure should mirror the evidence in Fig 1 a bit more (not including the original kinase inhibitor screen). The cells from Fig 1D (WT, AMPKa^{-/-}+Vec, +WT, +KD) should be used to test the requirement of AMPK for EGF-stimulated Akt. Its good to test the effects in other cell lines but if room is tight, 2B could go in supp data as this type of data did in Fig1. The authors should also test the effects of Compound C on EGF treatments. Again, Compound C is not very specific but is ok to use here because a basic observation is still being established (that AMPK influences EGF signaling to Akt).

We agree with reviewer that using the same cells (WT, AMPKa^{-/-} +Vec, +WT, +KD) as in Fig. 1 could strengthen our dataset in Fig. 2. Therefore, we tested the effect of AMPK restoration on AKT activation under EGF treatment. We observed that Akt activation with EGF treatment was also impaired in AMPKa^{-/-} MEFs. Restoration of AMPK, but not kinase-dead AMPKa, rescued the defect in Akt phosphorylation in AMPKa^{-/-} MEFs upon EGF treatment (Revised Fig. 2b). As reviewer's suggestion, we also tested compound C treatment on MDA-MB-231 cells in presence and absence of EGF and impaired EGF-induced Akt phosphorylation under compound C treatment was observed (Revised Supplementary Fig. 2d).

2-2. The phosphorylation of the EGFR and ERK should be shown throughout this figure to establish whether AMPK loss is having upstream or downstream effects on EGF signaling other than on Akt. This will help readers evaluate the validity of the proposed mechanism of AMPK-dependent effects on EGF-stimulated AKT activation.

As suggested, we evaluated EGFR and ERK phosphorylation in AMPK knockdown MDA-MB-231 cells under EGF treatment and found that AMPK knockdown did not impact on EGFR and ERK phosphorylation upon EGF treatment (Revised Fig 2a). Moreover, we could not detect any changes in phospho-EGFR and phospho-ERK level in either CaMKK β deficient or addback cells (Revised Fig 2c, e).

2-3. It's nice that the CAMKKb data consists of complimentary shRNA-knockdown and inhibitor experiments. However, for the shRNA knockdown experiment the shLUC, shCAMKKb, and shLKB1 should be done side by side in the same experiment and run on the same gel to make a single subfigure (instead of in separate figures 2c and sup 2C) Only in this way can the reader properly compare the effects of the two upstream kinases on AMPK signaling. Following this experiment, ideally, a second separate a CAMKKb addback rescue experiment should be included to strengthen the data on involvement of CAMKKb since this point is specifically included in the abstract.

We thank the reviewer for suggestion on putting shLUC, shCAMKKb, and shLKB1 all together for better comparison. However, we did not do it due to the gigantic datasets we have in this study and have presented them separately in main and supplementary data due to the limited space. We think the data we presented should be acceptable because each group was treated under same conditions and accompanied with control groups.

We agree with the reviewer that that a separate CAMKKb addback rescue experiment will further strengthen our result. As suggested, we have restored CAMKKb in CAMKKb deficient cells to monitor EGF signaling pathway. We observed that the impairment of phospho-Akt (T308 and S473) and two Akt downstream readouts (phospho-GSK3 β and phospho-PRAS40) was rescued upon CAMKKb restoration (Revised Fig. 2e).

Figure 3) This figure should include data on glucose deprivation and oxidative stress, should eliminate use of Compound C, and should make use of addback control cells.3-1. The authors specifically mention glucose deprivation and oxidative stress (in addition to hypoxia) in the abstract. Figures showing the effects on Akt ubiquitination in response to glucose deprivation and H2O2 should therefore be in the main figure. These should be done with knockdown/knockout cells with addback controls. Compound C should not be used in these experiments to implicate AMPK because it is too non-specific and we have moved beyond basic observation experiments.

3-2. Add an experiment with the AMPK-null MEFs (including addback controls as in 1d to strengthen this figure.

We thank reviewer for the constructive suggestion. Compound C treatment data have been removed as advised. We have followed reviewer's suggestions and examined Akt

ubiquitination in AMPK knockout and addback cells under all three stress conditions. We overexpressed Akt to boost the endogenous Akt ubiquitination signal in WT, AMPK null and AMPK addback groups. We found that Akt ubiquitination was impaired in AMPK knockout MEFs, but was rescued in AMPK addback group under hypoxia, glucose deprivation and H₂O₂ treatment (Revised Fig 3c-e).

3-3. Not clear why an Akt IP wasn't done instead of Ub IP (as they did in their 2012 paper)? At least in that case it could be shown that total Akt levels aren't changing in the IP but Ub levels are.

The approach in performing Ub IP followed by Akt detection could ensure the ubiquitination signal is specific to Akt. In any case, we have used both methods to validate the result for Akt ubiquitination.

3-4. Fig 3b is nice but how do we know that WT, K63R, and K48R Ub are expressed at similar levels in the cell? Differences in expression levels could also explain these observations. Transfected Ub levels need to somehow be shown.

As suggested, transfected His-Ub levels were detected using His antibody in Revised Fig. 3b.

3-5. AKT mislabeled? –In 3D and 3E is total Akt really from the IP? Seems to be mislabeled and is really from WCE.

We apologize for our mistakes. We have corrected the errors in revised Figure 3c and 3g.

3-6. In 3f, total Akt is overexposed. Akt is the main thing we're interested in. Why is it overexposed?? Need to see whether Akt levels are even across cell lines.

AS suggested, a lower exposure of Akt band has been shown in the revised Figure 3h.

Figure 4) 4f is a good experiment and the sequence of SKP2-S256, while differing from the main AMPK consensus motif, looks similar to some other AMPK substrates. But the constitutively active AMPK is a problematic reagent and experiments with addback controls are needed.

4-1. Add an experiment with the AMPK-null MEFs (including addback controls as in 1d) to back up shAMPK data and strengthen this figure.

To determine whether Skp2 S256 phosphorylation is regulated by AMPK, we restored AMPK α in AMPK α null MEFs and monitored Skp2 S256 phosphorylation under stresses. We found that add-back of WT AMPK rescued Skp2 S256 phosphorylation under EGF, glucose deprivation and hypoxia treatment (Revised Supplementary Fig. 4a).

4-2. Put the SKP2 sequence alignment figure from Rebuttal, Response #2 into main figure 4! Show alignment of SKP2-S256 and surrounding sequence with the consensus and secondary consensus AMPK motifs from Gwinn et al Mol Cell 2008 along the top (including position numbering). Importantly, add the well-established substrates Raptor S722 and ACC1/2 S79/80

which both also have an “S” at the -3 position instead of an R! Put these, along with ULK1 right under SKP2 sequence in the alignment. Get rid of NO synthase from this figure – doesn’t really help at all. Add references to this subfigure for each substrate. Directly address the differences in the SKP2 sequence compared to the optimal motif pointing out that an “S” at the -3 position is tolerated in other accepted substrates. Also explicitly point out the L is in the +5 position not the +4 position. Also, there is an acidic residue preceding the L at +3(D) and +4(E) position which is consistent with the +3D/E in the secondary motif listed in Gwinn et al. Readers in the signaling field will appreciate this and will be more likely to accept this as an AMPK substrate.

We thank the reviewer for detailed instructions that help strengthen our findings. As suggested, we have replaced rebuttal figure 4b in main figure with Revised Figure 4b and put previous Figure 4b into supplementary figures. A relevant statement was added in the discussion, as suggested.

4-3. Results with Constitutively Active mutant of AMPK are questionable. The constitutively active mutant used here is not sufficiently described in the methods but seems to be a truncated version of AMPK lacking its autoinhibitory domain. There are two problems here. A) Not only is this mutant lacking a big chunk of the endogenous alpha subunit, this mutant does not associate with the beta and gamma regulatory subunits of the full heterotrimeric AMPK complex. The other subunits are required for proper localization and also may affect interactions with substrates. In cells, the CA mutant will be diffusely mislocalized and even in in vitro assays the lack of beta and gamma subunits may allow the alpha subunit to phosphorylate sites it might not otherwise target in the WT complex. Either way, the truncated CA AMPK has the potential to yield misleading results. B) When used in the same experiment, the levels of CA and KD AMPK, which are very different sizes, are shown on separate panels making it impossible to judge their relative expression levels. CA and KD AMPK need to be at similar molar amounts for KD to serve as a proper control. Other papers have used WT AMPK successfully in cells and in vitro kinase assays. Why not use WT AMPK?

A) We used CA AMPK because it could fully activate AMPK and its downstream signaling and is extensively used in the field (REFs), although we could completely rule out the possibility that CA AMPK overexpression might result in Skp2 phosphorylation through mechanisms other than AMPK. As suggested, we also used WT AMPK heterotrimeric complex for in vitro kinase assay (Revised Fig. 4f) and showed that WT AMPK phosphorylates Skp2 at S256 in vitro.

B) We agree that similar molar amount for CA and KD AMPK s necessary for their comparison. Therefore, we have reblotted CA and KD in the same panel in revised Figure 4d.

4-4. An important point to be drawn from 4G is how the timing of SKP2-S256 phosphorylation correlates with the timing of AMPK-T172 phosphorylation (and ACC phosphorylation) plus AKT phosphorylation in response to these stimuli. Phospho-T172-AMPK, (maybe phospho-ACC), and Phospho-T308 and -S473-Akt should be shown in the WCE lysate for all of these so that we can see them in the same experiment and evaluate whether their timing makes sense.

We used the same time points for Fig. 1 and all pAkt and pAMPK as well as downstream effectors were shown in Fig. 1. The time point for hypoxia is 4 hours when both Akt and AMPK were activated. Time points for EGF, glucose deprivation and H₂O₂ treatment are 5 minutes, 1 hour and 1 hour, respectively.

4-5. The most current version of Scansite, Scansite4 (<https://scansite4.mit.edu/4.0/>) does not predict the same putative AMPK sites on SKP2 as listed here (other than S256). Also, it does not predict any AMPK sites on SKP2 until it is placed on the two lowest stringency settings (Low and Minimum). These differences will be confusing to readers and should be rectified some how. Also, it should be stated that scansite was used on the lowest stringency settings (which is fine given the sequence alignments with Raptor and ACC).

As suggested, the statement that “scansite was used on the lowest stringency settings” was added to the figure legend (Revised Supplementary Fig. 4c) to avoid misleading.

4-6. There are no data showing that S256A or D mutants affect Akt phosphorylation/signaling (e.g. S256A or D addbacks to SKP2-null MEFs and show phospho-Akt, phospho-PRAS40, phospho-GSK3) This is critical to complete the mechanistic link from AMPK to Akt that is the focus of this study as described in the title and abstract. (Note: an experiment with an Akt mutant that could not be Ubiquitinated by SKP2, and hence lost sensitivity to AMPK, would be the final piece of data needed to complete the pathway)

As suggested, we have restored Skp2 S256A or S256D mutants into Skp2 null MEFs, treated cells with hypoxia for 4 hours, and examined Akt signaling in those cells. We found that phospho-Akt T308, S473 and phospho-PRAS40 could only be rescued in S256D Skp2 addbacks but not S256A addbacks (Revised Supplementary Fig. 4b). We also introduced HA-AMPK α 1, Akt WT and its ubiquitination deficient mutant K8R into AMPK α null MEFs and examined Akt signaling under EGF treatment in those cells. We found that restoration of AMPK and WT Akt rescued defects in Akt and PRAS40 phosphorylation, but Akt ubiquitination deficient mutant K8R failed to do so (Revised Supplementary Fig. 4c). These data suggest that AMPK regulates Akt signaling through Akt ubiquitination.

Figures 5 and 6) While there is compelling in vivo data with the SKP2-S256A and D mutants. There is also a lack of data connecting the in vivo effects of SKP2 mutants to AKT in either Fig 5 or Fig 6.

In order to connect Skp2 mutant to Akt, we stained Formalin-Fixed, Paraffin-Embedded SubQ tumor sections of Skp2 S256A and S256D with Akt and pAkt T308 antibodies. We found that while S256D mutant was able to induce Akt T308 phosphorylation, Skp2 S256A phosphorylation deficient mutant failed to do so (Revised Supplementary Fig. 6d).

MINOR COMMENTS

1) Line up labels better on some figures! Recheck labeling on all figures.

We have corrected labels on all figures in the revised manuscript.

2) [page 6] maybe “the kinase inhibitor library” should be “a kinase inhibitor library”?

Thanks for pointing out the error. We have corrected it in the revised manuscript.

3) [page 6] Rephrase “Activation of PI3K, PDK1, PIM, and Src... were all required for Akt activation...” to say something like “PI3K, PDK1, PIM and Src inhibitors all blocked Akt activation” since some of these inhibitors are not extremely specific and alone can not be used to conclude those targets are the key (just being conservative here – I know there’s literature on each of these inhibitors/targets).

We agree with the reviewer, and corrections were made in revised manuscript.

4) [Fig 1D]. should show total endogenous AMPK levels to demonstrate KO (not just HA-tagged AMPK levels)

As suggested, total endogenous AMPK level has been shown in Revised Figure 1d.

5) [Fig 2e] Legend for 2e should say how long EGF treatment was.

EGF was treated for 5 minutes. We have included this information in the revised figure legend.

6) FIG 2] Mislabeling in legend. (d) is listed twice meaning last two subfigure legends are mislabeled.

We have corrected the errors in the revised manuscript.

7) There does appear to be some effect on Akt activation in shAMPK cells in response to IGF (its just that the figure seems to be overexposed). There even seems to be a bit of an effect on PDGF-stimulated AKT. Perhaps soften the language that AMPK is “dispensable for Akt phosphorylation and activation by IGF-1 and PDGF...”. This doesn’t seem to be completely true.

We agree with the reviewer and have changed the statement to “not significant”, since we did not further confirm our results for other growth factors except for EGF.

8) CAMKKb has been reported to directly phosphorylate T308-Akt directly. While the evidence for this is quite thin, the authors might want to very briefly address this in the discussion.

As suggested, we have included the reference and discussed the possibility that CAMKK β may directly phosphorylates Akt on T308.

Reference:

1. Mungai, P.T., *et al.* Hypoxia triggers AMPK activation through reactive oxygen species-mediated activation of calcium release-activated calcium channels. *Molecular and cellular biology* 31, 3531-3545 (2011).

Reviewers' comments:

Reviewer #2 (Remarks to the Author):

Overall, the authors have addressed most of the questions that were raised:

- a) the manuscript has been re-organized resulting in improved readability;
- b) some of the claims are now supported with additional data points (for example, additional blots with mTOR targets are presented in addition to blots examining pS6 status);
- c) language in the description of the results has been softened to better reflect the results.

Reviewer #4 (Remarks to the Author):

****PLEASE SEE ADDITIONAL "REVIEWER NEW COMMENTS" BELOW: ****

Reviewer #4 (Remarks to the Author):

SUMMARY: The authors propose an AMPK-dependent signaling pathway through which AKT can be activated by cellular stresses including hypoxia, glucose starvation, and ROS and also by the growth factor EGF. Specifically, this pathway consists of AMPK phosphorylating and activating the E3 Ubiquitin Ligase SKP2 which then ubiquitinates AKT to promote its activation (ubiquitination is nonproteolytic). Activation of AKT via this pathway is proposed to support cancer cell survival and tumor progression. This study builds off of the author's previous study (Chan et al., Cell 2012) which showed that SKP2 could ubiquitinate Akt promoting its activation and that this was required for full activation of Akt by EGF. Therefore the novel points about this Akt activation mechanism in this manuscript include that AMPK phosphorylates and activates SKP2 and that this mediates AKT activation by EGF and the cellular stresses listed above. This study contains some promising data but currently falls short of connecting all the dots to support their proposed mechanism. For instance, there is no data showing that the S256A-SKP2 mutant affects Akt activation/signaling and there is no data showing that an AKT mutant that can't be ubiquitinated by SKP2 is resistant to the effects of AMPK. Some additional controls and complementary experiments are needed to strengthen the mechanistic figures. Detailed comments are listed below:

AUTHOR RESPONSE: Thanks for recognizing the novelty of our study and for offering constructive comments that help further strengthen our study. We have taken the reviewer's comments by heart and addressed all these concerns with relevant experiments (see below).

GENERAL COMMENTS:

1) AMPK-inhibition/deficiency is likely to lead to increased activation of mTORC1 (due to relief of AMPK-dependent TSC2 and Raptor inhibitory phosphorylation). Increased mTORC1 activation could in turn suppress Akt activation through the multiple mTORC1-dependent negative feedback loops that inhibit PI3K-mTORC2-Akt signaling. While the authors address this issue to some extent in the text and Supp Fig 8 they need to do more to rule this pathway out. The rapamycin treatments show that mTORC1 signaling can affect Akt activation in these cells. Monitoring S6 phosphorylation is generally not sufficient to determine effects on mTORC1 – while the phospho-S6 antibody used here is not listed in the methods, the most common site monitored, S235/236 can be phosphorylated by RSK-downstream of ERK as well as S6K downstream of mTORC1. A better and direct readout of mTORC1 signaling is phospho-T389-S6K. The effects of shAMPK knockdown and treatments on phospho-T389-S6K and total S6K should be shown in Sup Fig 8 instead of S6. It could also be shown whether there are any changes in the mTORC1 feedback substrate Grb10-S476 (antibody from cell signaling) which would affect RTK-Akt signaling. One problem is that since mTORC1 activity is not fully sensitive to rapamycin and since mTORC1 directly phosphorylates Grb10, rapamycin may not fully rescue phospho-Akt in AMPK-deficient cells. Admittedly, this can be a difficult pathway to rule out since there is no specific catalytic inhibitor of mTORC1 that would not also hit mTORC2. But the authors need to more clearly rule out this

pathway for readers.

AUTHOR RESPONSE: We agree with reviewer that we need more readouts to rule out mTORC1 pathway in order to claim the importance of AMPK-Skp2 axis in Akt activation. We have added the information for S6 phosphorylation site (S235/236) in the method section (IP and IB). In supplemental Fig. 8a, rapamycin seemed to inhibit hypoxia-induced phospho-S6 very efficiently. Therefore, we believe that mTORC1 is sensitive to rapamycin, which further triggers positive feedback loop to induce Akt activation. To better monitor mTORC1 activity, we used both phospho-S6K(T389) and phospho-Grb10(S476) as readouts. Results show that mTORC1 activation was not enhanced in AMPK deficient MDA231 cells under EGF, glucose deprivation and H2O2 treatment (Revised Supplementary Fig. 8b-c). However, we detected impaired Akt activation in the same cell line under the same condition. Therefore, we rule out the possibility that AMPK mediates Akt activation through upregulating mTORC1.

REVIEWER NEW COMMENT: The data on mTORC1 still falls short of properly addressing the relative role of mTORC1 in AMPK-dependent Akt activation. Addressing this point is important because it will likely be the first mechanism that pops into the minds of readers familiar with the AMPK and mTOR literature when they see the effects of AMPK loss on Akt presented here. In fact, in my opinion this point should be addressed earlier on in the paper and not saved for the discussion and Sup Fig 8.

The point of these experiments is not to show that AMPK doesn't affect Akt at all through mTORC1-dependent mechanisms – that would contradict a good deal of published data. The point is simply to show that AMPK doesn't ONLY affect Akt through mTORC1, leaving the window open for additional mechanisms. This would be addressed by demonstrating that AMPK loss still affects Akt even when mTORC1 is inhibited (e.g. with rapamycin). This helps to rule out that AMPK loss leads to decreased phospho-Akt only because AMPK-dependent inhibition of mTORC1 is relieved leading to increased mTORC1-dependent signaling and negative feedback inhibition of PI3K-Akt signaling.

RELATED NOTE 1: In the discussion, the text on glucose deprivation directly contradicts the data in Sup Fig 8c. The text states, "...we found that neither phosphorylation of S6K and Grb10, mTOR downstream effectors, was induced under glucose deprivation...". However, the data in Sup Fig 8C show phospho-T389-S6K being induced under the GD+ condition.

RELATED NOTE 2. This statement in the discussion is too strong "These data suggest that AMPK does not impact mTORC1 activity in response to stresses and EGF." AMPK knockdown does appear to increase basal levels of mTORC1 signaling (phospho-S6K) in several experiments where basal levels of phospho-S6K can be discerned. For example, in Sup Fig8b, phospho-S6K is higher in unstimulated shAMPK cells compared to unstimulated shLUC. Also in Sup Fig8D the phospho-S6K is overexposed but seems to be higher in shAMPK cells.). Even in Sup Fig8A, phospho-S6 is higher in shAMPK cells treated with rapamycin compared to shLUC cells treated with rapamycin. Pre-existing higher basal levels of mTORC1 signaling in shAMPK cells could suppress the maximum amount AKT can be stimulated in these cells.

Each experiment currently shown in Sup Fig 8 should include the following:

- a. Rapamycin treatments (needed for experiments with EGF, Glucose deprivation, and H2O2 - Sup Fig8B-D).
- b. Readouts for Akt signaling including Phospho-T308-Akt, phospho-S473-Akt, Total Akt, phospho-PRAS40, and total PRAS40 (need to be shown for EGF, glucose deprivation, and H2O2 experiments - Sup Fig 8b-d).
- c. Readouts for AMPK signaling including total levels of AMPK to confirm level of knockdown in

each experiment and Phospho-ACC and total ACC (needed in Sup FigA-D).

2) Compound C should not be used for the more detailed mechanistic experiments. The literature indicates that Compound C is far from being a specific inhibitor of AMPK. Some publications even suggest Compound C can inhibit mTOR-Akt signaling in an AMPK-independent manner. A kinase inhibitor database (<http://www.kinase-screen.mrc.ac.uk/kinase-inhibitors> <https://urldefense.proofpoint.com/v2/url?u=http-3A__www.kinase-2Dscreen.mrc.ac.uk_kinase-2Dinhibitors&d=DwMGAg&c=1jSOPko_E6DLthr5DiSb_42CaaM9nu9QAD4dem7yWs&r=cotLUCHF_IQYKz7sBxqp6sDh4UjpetQaj4cPFbRp6EQ&m=LMg1s3cU5a5B6o8LE0CbI9jnzKME1C56H2xmjrlgYW0&s=cqgoOfjAo6GSQk78tSzkOdOstU5C6AOpz3v6HO8qXLA&e=>>) indicates that a couple dozen kinases are inhibited by Compound C to a similar or greater extent than AMPK including kinases that could affect Akt signaling. The use of Compound C should be reserved for the initial screen and making the most basic observations (Figures 1 and 2). It seems quite useless for later mechanistic experiments where the authors are attempting to establish stronger conclusions about the specific role of AMPK (Fig 3, Sup Fig 3, and Fig 4) The Compound C data from these later figures should be removed because at best its weak support for the thesis and at worst potentially misleading. (with the exception of Fig 4f in which its use is ok since it's an in vitro assay).

AUTHOR RESPONSE: We agree with reviewer that Compound C is not entirely specific for AMPK targeting. We have removed compound C data for mechanistic study, as suggested.

REVIEWER NEW COMMENT: OK

3) More experiments should make use of AMPK addback controls in AMPK-deficient cells. Only one experiment makes use of the AMPK α -null MEFs with addbacks (Fig 1d) – why aren't these used in other experiments. The shAMPK knockdown cells are more commonly used but there is no addback control for these. Either use the AMPK α -null MEF addbacks for more key experiments or make the addbacks for the shAMPK knockdown cells.

AUTHOR RESPONSE: As suggested, we have restored WT and kinase-dead AMPK in AMPK-null MEFs and examined Akt activation upon various stress conditions (Glucose deprivation and H₂O₂). We observed that restoration of WT AMPK, but not kinase-dead AMPK, rescued Akt activation under glucose deprivation and H₂O₂ treatment (Revised Fig. 1d, e).

REVIEWER NEW COMMENT: OK

4) Readout of cellular Akt signaling should include one or two consistent substrates across all figures for which the point is to evaluate the effects on Akt signaling. Phospho-T308 and -S473 are shown in every figure of this type which is good. But not all figures include an Akt substrate to help interpret the effects on Akt signaling. In one figure both GSK3 and PRAS40 are shown, in some figure only GSK3 is shown, and in other figures no substrate is shown. Ideally, both PRAS40 and GSK3 phosphorylation status would be consistently shown in every figure in which the effects of a treatment on Akt activity/signaling must be interpreted. If the authors only want to show one substrate, PRAS40 may be best since it is a more specific target of Akt compared to GSK3 (which can be phosphorylated on the same sites by the related kinases S6K and RSK that might also be affected by the treatments used here). The same readouts for Akt should be shown in every figure.

AUTHOR RESPONSE: We agree with the reviewer that two readouts are indispensable in monitoring Akt activation under particular conditions. We have repeated the experiments and

included both phospho-GSK3b and phospho-PRAS40 reflecting Akt activation along with phospho-Akt T308 and S473 under all conditions-hypoxia, glucose deprivation, H₂O₂ and EGF in the results, and the similar results were obtained by examining phospho-GSK3b or phospho-PRAS40 signal. At least one consistent readout was shown thereafter.

REVIEWER NEW COMMENT: This isn't completely true. A readout of Akt signaling (phospho-PRAS40 and total PRAS40) is missing from Fig 1C and 1D. Why wouldn't you make Figures 1C, 1D, and 1E all uniform with the same blots in the same order?? Phospho-PRAS40 is also missing from Fig 2A, 2C, 2D, and 2F, while it is shown in 2B and 2E. Show it in every figure so that they have a readout in common! You can't just show phospho-GSK3 in some subfigures and phospho-PRAS40 in others for no good reason. Be consistent. There's other figures where only phospho-Akt or phospho-GSK3 is shown but I'm not going to list them all. Again, bottom line is there should be the same readouts shown consistently across figures so readers can properly compare across different cell lines, treatments, and types of experiments.

5) Fix misaligned labeling of figures. For example, in 1D, 2A, 4A, 4D, 4E, Sup4A, etc.

AUTHOR RESPONSE: We thank the reviewer for pointing our mistakes. We have corrected these errors in revised Figures.

REVIEWER NEW COMMENT: OK

FIGURE-SPECIFIC COMMENTS:

Figure 1) Basic observation about requirement of AMPK for hypoxia, glucose starvation, ROS-induced Akt is convincing but the evidence should be strengthened. The evidence for the basic observation that stress-induced Akt activation is diminished in AMPK-deficient cells is fairly convincing since complementary data from an inhibitor, shAMPK-knockdown cells, and AMPK α null MEFs is shown. 1D is a good experiment with the proper addback controls. However, Figure 1 as a whole should be strengthened by including the addback controls (WT and KD AMPK addback) used in 1D as controls in 1E (glucose starvation) and 1F (ROS) since these are distinct stimuli being tested for the first time. Ideally, addback rescue controls would have been included for the shAMPK α 1 cells also. Compound C should be used with caution as it is not a very specific inhibitor but it is fine here when used simply to establish the basic observation that AMPK appears to be required for full Akt activation in these settings. Compound C treatments after glucose deprivation and H₂O₂ treatment should also be shown in Figure 1 for uniformity.

AUTHOR RESPONSE: We thank the reviewer for the thoughtful suggestions. As suggested, we have restored WT and kinase-dead AMPK into AMPK-null MEFs and examined Akt activation under glucose deprivation and H₂O₂. We found that phospho-Akt T308 and S473 were rescued by WT but not kinase-dead AMPK addbacks upon glucose deprivation and H₂O₂ treatments (Revised Fig. 1d, e). We also treated MDA-MB-231 cells with Compound C under glucose deprivation and H₂O₂ and observed impaired Akt phosphorylation under these conditions (Revised Supplementary Fig. 1i, j).

REVIEWER NEW COMMENT: OK

Figure 2) The data supporting the basic observation about requirement of AMPK for EGF signaling should be strengthened. 2-1. This figure should mirror the evidence in Fig 1 a bit more (not including the original kinase inhibitor screen). The cells from Fig 1D (WT, AMPK α -/-+Vec, +WT, +KD) should be used to test the requirement of AMPK for EGF-stimulated Akt. It's good to test the effects in other cell lines but if room is tight, 2B could go in supp data as this type of data did in

Fig1. The authors should also test the effects of Compound C on EGF treatments. Again, Compound C is not very specific but is ok to use here because a basic observation is still being established (that AMPK influences EGF signaling to Akt).

AUTHOR RESPONSE: We agree with reviewer that using the same cells (WT, AMPK α -/- +Vec, +WT, +KD) as in Fig. 1 could strengthen our dataset in Fig. 2. Therefore, we tested the effect of AMPK restoration on AKT activation under EGF treatment. We observed that Akt activation with EGF treatment was also impaired in AMPK α -/- MEFs. Restoration of AMPK, but not kinase-dead AMPK α , rescued the defect in Akt phosphorylation in AMPK α -/- MEFs upon EGF treatment (Revised Fig. 2b). As reviewer's suggestion, we also tested compound C treatment on MDA-MB-231 cells in presence and absence of EGF and impaired EGF-induced Akt phosphorylation under compound C treatment was observed (Revised Supplementary Fig. 2d).

REVIEWER NEW COMMENT: OK. But there's no description of how AMPK Addback cells were made.

2-2. The phosphorylation of the EGFR and ERK should be shown throughout this figure to establish whether AMPK loss is having upstream or downstream effects on EGF signaling other than on Akt. This will help readers evaluate the validity of the proposed mechanism of AMPK-dependent effects on EGF-stimulated AKT activation.

AUTHOR RESPONSE: As suggested, we evaluated EGFR and ERK phosphorylation in AMPK knockdown MDA-MB-231 cells under EGF treatment and found that AMPK knockdown did not impact on EGFR and ERK phosphorylation upon EGF treatment (Revised Fig 2a). Moreover, we could not detect any changes in phospho-EGFR and phospho-ERK level in either CaMKK β deficient or addback cells (Revised Fig 2c, e).

REVIEWER NEW COMMENT: Need to show EGFR and ERK phosphorylation in the only addback experiment in this figure, Fig2B.

2-3. It's nice that the CAMKK β data consists of complimentary shRNA-knockdown and inhibitor experiments. However, for the shRNA knockdown experiment the shLUC, shCAMKK β , and shLKB1 should be done side by side in the same experiment and run on the same gel to make a single subfigure (instead of in separate figures 2c and sup 2C) Only in this way can the reader properly compare the effects of the two upstream kinases on AMPK signaling. Following this experiment, ideally, a second separate CAMKK β addback rescue experiment should be included to strengthen the data on involvement of CAMKK β since this point is specifically included in the abstract.

AUTHOR RESPONSE: We thank the reviewer for suggestion on putting shLUC, shCAMKK β , and shLKB1 all together for better comparison. However, we did not do it due to the gigantic datasets we have in this study and have presented them separately in main and supplementary data due to the limited space. We think the data we presented should be acceptable because each group was treated under same conditions and accompanied with control groups.

REVIEWER NEW COMMENT: If shCAMKK β and shLKB1 knockdowns are going to be shown in separate experiments then at the very least show the exact same blots/readouts. For instance, why would you show phospho-ACC in Fig 2C but phospho-AMPK in SupFig 2E? Why isn't a readout for Akt signaling shown in SupFig 2E? Why aren't p-EGFR and pERK shown in SupFig 2E? Just show the same exact set of blots for Fig2C and SupFig 2E.

AUTHOR RESPONSE: We agree with the reviewer that that a separate CAMKK β addback rescue experiment will further strengthen our result. As suggested, we have restored CAMKK β in CAMKK β deficient cells to monitor EGF signaling pathway. We observed that the impairment of phospho-Akt (T308 and S473) and two Akt downstream readouts (phospho-GSK3 β and phospho-PRAS40) was rescued upon CAMKK β restoration (Revised Fig. 2e).

REVIEWER NEW COMMENT: OK. But must describe in methods how CAMKKB addback cells were made.

Figure 3) This figure should include data on glucose deprivation and oxidative stress, should eliminate use of Compound C, and should make use of addback control cells.3-1. The authors specifically mention glucose deprivation and oxidative stress (in addition to hypoxia) in the abstract. Figures showing the effects on Akt ubiquitination in response to glucose deprivation and H2O2 should therefore be in the main figure. These should be done with knockdown/knockout cells with addback controls. Compound C should not be used in these experiments to implicate AMPK because it is too non-specific and we have moved beyond basic observation experiments.

3-2. Add an experiment with the AMPK-null MEFs (including addback controls as in 1d to strengthen this figure.

AUTHOR RESPONSE: We thank reviewer for the constructive suggestion. Compound C treatment data have been removed as advised. We have followed reviewer's suggestions and examined Akt ubiquitination in AMPK knockout and addback cells under all three stress conditions. We overexpressed Akt to boost the endogenous Akt ubiquitination signal in WT, AMPK null and AMPK addback groups. We found that Akt ubiquitination was impaired in AMPK knockout MEFs, but was rescued in AMPK addback group under hypoxia, glucose deprivation and H2O2 treatment (Revised Fig 3c-e).

REVIEWER NEW COMMENT: OK

3-3. Not clear why an Akt IP wasn't done instead of Ub IP (as they did in their 2012 paper)? At least in that case it could be shown that total Akt levels aren't changing in the IP but Ub levels are.

AUTHOR RESPONSE: The approach in performing Ub IP followed by Akt detection could ensure the ubiquitination signal is specific to Akt. In any case, we have used both methods to validate the result for Akt ubiquitination.

REVIEWER NEW COMMENT: OK

3-4. Fig 3b is nice but how do we know that WT, K63R, and K48R Ub are expressed at similar levels in the cell? Differences in expression levels could also explain these observations. Transfected Ub levels need to somehow be shown.

AUTHOR RESPONSE: As suggested, transfected His-Ub levels were detected using His antibody in Revised Fig. 3b.

REVIEWER NEW COMMENT: OK

3-5. AKT mislabeled? –In 3D and 3E is total Akt really from the IP? Seems to be mislabeled and is really from WCE.

AUTHOR RESPONSE: We apologize for our mistakes. We have corrected the errors in revised Figure 3c and 3g.

REVIEWER NEW COMMENT: OK

3-6. In 3f, total Akt is overexposed. Akt is the main thing we're interested in. Why is it overexposed?? Need to see whether Akt levels are even across cell lines.

AUTHOR RESPONSE: AS suggested, a lower exposure of Akt band has been shown in the revised Figure 3h.

REVIEWER NEW COMMENT: OK

Figure 4) 4f is a good experiment and the sequence of SKP2-S256, while differing from the main AMPK consensus motif, looks similar to some other AMPK substrates. But the constitutively active AMPK is a problematic reagent and experiments with addback controls are needed.

4-1. Add an experiment with the AMPK-null MEFs (including addback controls as in 1d) to back up shAMPK data and strengthen this figure.

AUTHOR RESPONSE: To determine whether Skp2 S256 phosphorylation is regulated by AMPK, we restored AMPK α in AMPK α null MEFs and monitored Skp2 S256 phosphorylation under stresses. We found that add-back of WT AMPK rescued Skp2 S256 phosphorylation under EGF, glucose deprivation and hypoxia treatment (Revised Supplementary Fig. 4a).

REVIEWER NEW COMMENT: OK

4-2. Put the SKP2 sequence alignment figure from Rebuttal, Response #2 into main figure 4! Show alignment of SKP2-S256 and surrounding sequence with the consensus and secondary consensus AMPK motifs from Gwinn et al Mol Cell 2008 along the top (including position numbering). Importantly, add the well-established substrates Raptor S722 and ACC1/2 S79/80 which both also have an "S" at the -3 position instead of an R! Put these, along with ULK1 right under SKP2 sequence in the alignment. Get rid of NO synthase from this figure – doesn't really help at all. Add references to this subfigure for each substrate. Directly address the differences in the SKP2 sequence compared to the optimal motif pointing out that an "S" at the -3 position is tolerated in other accepted substrates. Also explicitly point out the L is in the +5 position not the +4 position. Also, there is an acidic residue preceding the L at +3(D) and +4(E) position which is consistent with the +3D/E in the secondary motif listed in Gwinn et al. Readers in the signaling field will appreciate this and will be more likely to accept this as an AMPK substrate.

AUTHOR RESPONSE: We thank the reviewer for detailed instructions that help strengthen our findings. As suggested, we have replaced rebuttal figure 4b in main figure with Revised Figure 4b and put previous Figure 4b into supplementary figures. A relevant statement was added in the discussion, as suggested.

REVIEWER NEW COMMENT: OK

4-3. Results with Constitutively Active mutant of AMPK are questionable. The constitutively active mutant used here is not sufficiently described in the methods but seems to be a truncated version of AMPK lacking its autoinhibitory domain. There are two problems here. A) Not only is this mutant lacking a big chunk of the endogenous alpha subunit, this mutant does not associate with the beta and gamma regulatory subunits of the full heterotrimeric AMPK complex. The other subunits are required for proper localization and also may affect interactions with substrates. In cells, the CA mutant will be diffusely mislocalized and even in in vitro assays the lack of beta and gamma subunits may allow the alpha subunit to phosphorylate sites it might not otherwise target in the WT complex. Either way, the truncated CA AMPK has the potential to yield misleading results. B) When used in the same experiment, the levels of CA and KD AMPK, which are very different sizes, are shown on separate panels making it impossible to judge their relative expression levels. CA and KD AMPK need to be at similar molar amounts for KD to serve as a proper control. Other papers have used WT AMPK successfully in cells and in vitro kinase assays. Why not use WT AMPK?

AUTHOR RESPONSE: A) We used CA AMPK because it could fully activate AMPK and its downstream signaling and is extensively used in the field (REFs), although we could completely rule out the possibility that CA AMPK overexpression might result in Skp2 phosphorylation through mechanisms other than AMPK. As suggested, we also used WT AMPK heterotrimeric complex for in vitro kinase assay (Revised Fig. 4f) and showed that WT AMPK phosphorylates Skp2 at S256 in vitro.

REVIEWER NEW COMMENT: OK. But need to describe the AMPKa1b1g1 complex used here better in methods. Where did it come from? How was it purified?

AUTHOR RESPONSE: B) We agree that similar molar amount for CA and KD AMPK s necessary for their comparison. Therefore, we have reblotted CA and KD in the same panel in revised Figure 4d.

REVIEWER NEW COMMENT: CA is expressed ~4X higher (molar) or more than KD? Not to mention a full length KD construct is being compared to a truncated CA construct. Not the best control.

4-4. An important point to be drawn from 4G is how the timing of SKP2-S256 phosphorylation correlates with the timing of AMPK-T172 phosphorylation (and ACC phosphorylation) plus AKT phosphorylation in response to these stimuli. Phospho-T172-AMPK, (maybe phospho- ACC), and Phospho-T308 and -S473-Akt should be shown in the WCE lysate for all of these so that we can see them in the same experiment and evaluate whether their timing makes sense.

AUTHOR RESPONSE: We used the same time points for Fig. 1 and all pAkt and pAMPK as well as downstream effectors were shown in Fig. 1. The time point for hypoxia is 4 hours when both Akt and AMPK were activated. Time points for EGF, glucose deprivation and H2O2 treatment are 5 minutes, 1 hour and 1 hour, respectively.

REVIEWER NEW COMMENT: Then there seems to be a disconnect between Skp2 phosphorylation and Akt phosphorylation in two ways. First, I would point out that in response to H2O2 the increase in phospho-Akt (at 1h) precedes any increase in phospho-Skp2 (at 2h) by a whole hour. How could Skp2 phosphorylation mediate the increase in phospho-Akt if Akt phosphorylation precedes Skp2 phosphorylation? Doesn't make sense. On the flipside of this point, for EGF and glucose deprivation, phospho-Skp2 decreases at later time points while AMPK signaling and phospho-Akt are still going up (for EGF, Fig 2A vs. Fig 4G; for GD, Fig 1G vs. Fig 4G). Again, it's hard to see how Skp2 phosphorylation is mediating Akt phosphorylation when they are going in opposite directions. I can't tell if something similar is happening with hypoxia because an 8h time point isn't shown in Fig 1B.

4-5. The most current version of Scansite, Scansite4 (<https://scansite4.mit.edu/4.0/>) does not predict the same putative AMPK sites on SKP2 as listed here (other than S256). Also, it does not predict any AMPK sites on SKP2 until it is place on the two lowest stringency settings (Low and Minimum). These differences will be confusing to readers and should be rectified some how. Also, it should be stated that scansite was used on the lowest stringency settings (which is fine given the sequence alignments with Raptor and ACC).

AUTHOR RESPONSE: As suggested, the statement that "scansite was used on the lowest stringency settings" was added to the figure legend (Revised Supplementary Fig. 4c) to avoid misleading.

REVIEWER NEW COMMENT: OK

4-6. There are no data showing that S256A or D mutants affect Akt phosphorylation/signaling (e.g. S256A or D addbacks to SKP2-null MEFs and show phospho-Akt, phospho-PRAS40, phospho-GSK3) This is critical to complete the mechanistic link from AMPK to Akt that is the focus of this

study as described in the title and abstract. (Note: an experiment with an Akt mutant that could not be Ubiquitinated by SKP2, and hence lost sensitivity to AMPK, would be the final piece of data needed to complete the pathway)

AUTHOR RESPONSE: As suggested, we have restored Skp2 S256A or S256D mutants into Skp2 null MEFs, treated cells with hypoxia for 4 hours, and examined Akt signaling in those cells. We found that phospho-Akt T308, S473 and phospho-PRAS40 could only be rescued in S256D Skp2 addbacks but not S256A addbacks (Revised Supplementary Fig. 4b). We also introduced HA-AMPK α 1, Akt WT and its ubiquitination deficient mutant K8R into AMPK α null MEFs and examined Akt signaling under EGF treatment in those cells. We found that restoration of AMPK and WT Akt rescued defects in Akt and PRAS40 phosphorylation, but Akt ubiquitination deficient mutant K8R failed to do so (Revised Supplementary Fig. 4c). These data suggest that AMPK regulates Akt signaling through Akt ubiquitination.

REVIEWER NEW COMMENT: OK

Figures 5 and 6) While there is compelling in vivo data with the SKP2-S256A and D mutants. There is also a lack of data connecting the in vivo effects of SKP2 mutants to AKT in either Fig 5 or Fig 6.

AUTHOR RESPONSE: In order to connect Skp2 mutant to Akt, we stained Formalin-Fixed, Paraffin-Embedded SubQ tumor sections of Skp2 S256A and S256D with Akt and pAkt T308 antibodies. We found that while S256D mutant was able to induce Akt T308 phosphorylation, Skp2 S256A phosphorylation deficient mutant failed to do so (Revised Supplementary Fig. 6d).

REVIEWER NEW COMMENT: OK

MINOR COMMENTS

1) Line up labels better on some figures! Recheck labeling on all figures.

AUTHOR RESPONSE: We have corrected labels on all figures in the revised manuscript.

REVIEWER NEW COMMENT: Closely look at each figure. There are still some misalignments of labels.

2) [page 6] maybe "the kinase inhibitor library" should be "a kinase inhibitor library"?

AUTHOR RESPONSE: Thanks for pointing out the error. We have corrected it in the revised manuscript.

REVIEWER NEW COMMENT: OK

3) [page 6] Rephrase "Activation of PI3K, PDK1, PIM, and Src... were all required for Akt activation..." to say something like "PI3K, PDK1, PIM and Src inhibitors all blocked Akt activation" since some of these inhibitors are not extremely specific and alone can not be used to conclude those targets are the key (just being conservative here – I know there's literature on each of these inhibitors/targets).

AUTHOR RESPONSE: We agree with the reviewer, and corrections were made in revised manuscript.

REVIEWER NEW COMMENT: OK

4) [Fig 1D]. should show total endogenous AMPK levels to demonstrate KO (not just HA-tagged AMPK levels)

AUTHOR RESPONSE: As suggested, total endogenous AMPK level has been shown in Revised Figure 1d.

REVIEWER NEW COMMENT: OK

5) [Fig 2e] Legend for 2e should say how long EGF treatment was.

AUTHOR RESPONSE: EGF was treated for 5 minutes. We have included this information in the revised figure legend.

REVIEWER NEW COMMENT: OK

6) FIG 2] Mislabeling in legend. (d) is listed twice meaning last two subfigure legends are mislabeled.

AUTHOR RESPONSE: We have corrected the errors in the revised manuscript.

REVIEWER NEW COMMENT: OK

7) There does appear to be some effect on Akt activation in shAMPK α cells in response to IGF (its just that the figure seems to be overexposed). There even seems to be a bit of an effect on PDGF-stimulated AKT. Perhaps soften the language that AMPK is "dispensable for Akt phosphorylation and activation by IGF-1 and PDGF...". This doesn't seem to be completely true.

AUTHOR RESPONSE: We agree with the reviewer and have changed the statement to "not significant", since we did not further confirm our results for other growth factors except for EGF.

REVIEWER NEW COMMENT: Maybe change to something else. "not significant" should be reserved for referring to statistical significance.

8) CAMKK β has been reported to directly phosphorylate T308-Akt directly. While the evidence for this is quite thin, the authors might want to very briefly address this in the discussion.

AUTHOR RESPONSE: As suggested, we have included the reference and discussed the possibility that CAMKK β may directly phosphorylates Akt on T308.

REVIEWER NEW COMMENT: OK

REVIEWER NEW COMMENT: Additional Minor Comments:

1) Please include the manufacturers catalogue number for reagents when applicable in methods section including for antibodies. This is standard practice these days.

2) For some "phospho-" antibodies in methods it says "phosphor". Please change all to "phospho-" in methods and throughout text for consistency.

3) Clearly label and state in legend for ALL main and supp figures which specific Skp2 and AMPK α shRNAs are being used since there are two for each. For instance, which AMPK shRNA is used in Fig 1b (shAMPK α 1-1 or 1-2?

4) The title of the paper doesn't fully reflect the contents of the paper – there's no mention of stress induced Akt activation.

Response to reviewers' comment

We would like to thank all the reviewers for recognizing the innovation and importance of our study and for providing constructive comments that in turn help strengthen our study tremendously. We have provided point-by-point answers to address all the reviewers' comments. Importantly, we have provided compelling new experimental data to address these concerns and hope the editor and reviewers will endorse the publication of this work at Nature Communication.

The point by point responses to the reviewers' comments are as followed:

Reviewer #1 (Remarks to the Author):

AUTHOR RESPONSE: The reviewer 1 has accepted our manuscript in the first submission.

Reviewer #2 (Remarks to the Author):

Overall, the authors have addressed most of the questions that were raised: a) the manuscript has been re-organized resulting in improved readability; b) some of the claims are now supported with additional data points (for example, additional blots with mTOR targets are presented in addition to blots examining pS6 status); c) language in the description of the results has been softened to better reflect the results.

AUTHOR RESPONSE: We thank the reviewer for recognizing the novelty and importance of our study and for accepting our revision.

Reviewer #4 (Remarks to the Author):

SUMMARY: The authors propose an AMPK-dependent signaling pathway through which AKT can be activated by cellular stresses including hypoxia, glucose starvation, and ROS and also by the growth factor EGF. Specifically, this pathway consists of AMPK phosphorylating and activating the E3 Ubiquitin Ligase SKP2 which then ubiquitinates AKT to promote its activation (ubiquitination is nonproteolytic). Activation of AKT via this pathway is proposed to support cancer cell survival and tumor progression. This study builds off of the author's previous study (Chan et al., Cell 2012) which showed that SKP2 could ubiquitinate Akt promoting its activation and that this was required for full activation of Akt by EGF. Therefore, the novel points about this Akt activation mechanism in this manuscript include that AMPK phosphorylates and activates SKP2 and that this mediates AKT activation by EGF and the cellular stresses listed above. This study contains some promising data but currently falls short of connecting all the dots to support their proposed mechanism. For instance, there is no

data showing that the S256A-SKP2 mutant affects Akt activation/signaling and there is no data showing that an AKT mutant that can't be ubiquitinated by SKP2 is resistant to the effects of AMPK. Some additional controls and complementary experiments are needed to strengthen the mechanistic figures. Detailed comments are listed below:

AUTHOR RESPONSE: We thank the reviewer for providing further constructive comments that help improve our manuscript tremendously. To further strengthen the proposed mechanism and to address reviewer's two major concerns, we have added two additional sets of the data:

1. In the **revised Supplementary Fig. 5b-e**, we restored WT and S256A mutant Skp2 into Skp2 null MEFs and treated cells with hypoxia, glucose deprivation, H₂O₂ or EGF. We found that WT Skp2 promoted Akt and PRAS40 phosphorylation but S256A-Skp2 failed to do so under all these circumstances. These data suggest Skp2 S256 phosphorylation is critical for Akt signaling activation under these treatments.

2. In order to investigate whether the Akt mutant that is not ubiquitinated by Skp2 is resistant to the action of AMPK, we restored AMPK along with WT HA-Akt or its ubiquitination-deficient mutant (HA-Akt K8R) into AMPK null MEFs. We then treated these cells with hypoxia, glucose deprivation, H₂O₂, or EGF and immunoprecipitated with HA antibody for examining Akt phosphorylation. We found that combined restoration of AMPK and WT Akt promoted Akt T308 phosphorylation. However, Akt alone and Akt K8R mutant along with AMPK failed to rescue Akt T308 phosphorylation (**revised Supplementary Fig.5 f-i**). These data suggest that Akt ubiquitination is responsible for AMPK-mediated Akt signaling activation.

GENERAL COMMENTS:

1) The data on mTORC1 still falls short of properly addressing the relative role of mTORC1 in AMPK-dependent Akt activation. Addressing this point is important because it will likely be the first mechanism that pops into the minds of readers familiar with the AMPK and mTOR literature when they see the effects of AMPK loss on Akt presented here. In fact, in my opinion this point should be addressed earlier on in the paper and not saved for the discussion and Sup Fig 8.

The point of these experiments is not to show that AMPK doesn't affect Akt at all through mTORC1-dependent mechanisms – that would contradict a good deal of published data. The point is simply to show that AMPK doesn't ONLY affect Akt through mTORC1, leaving the window open for additional mechanisms. This would be addressed by demonstrating that AMPK loss still affects Akt even when mTORC1 is inhibited (e.g. with rapamycin). This helps to rule out that AMPK loss leads to decreased phospho-Akt only because AMPK-dependent inhibition of mTORC1 is relieved leading to increased mTORC1-dependent signaling and negative feedback inhibition of PI3K-Akt signaling.

RELATED NOTE 1: In the discussion, the text on glucose deprivation directly contradicts the data in Sup Fig 8c. The text states, "...we found that neither

phosphorylation of S6K and Grb10, mTOR downstream effectors, was induced under glucose deprivation...". However, the data in Sup Fig 8C show phospho-T389-S6K being induced under the GD+ condition.

RELATED NOTE 2. This statement in the discussion is too strong "These data suggest that AMPK does not impact mTORC1 activity in response to stresses and EGF." AMPK knockdown does appear to increase basal levels of mTORC1 signaling (phospho-S6K) in several experiments where basal levels of phospho-S6K can be discerned. For example, in Sup Fig. 8b, phospho-S6K is higher in unstimulated shAMPK cells compared to unstimulated shLUC. Also in Sup Fig. 8D the phospho-S6K is overexposed but seems to be higher in shAMPK cells.). Even in Sup Fig. 8A, phospho-S6 is higher in shAMPK cells treated with rapamycin compared to shLUC cells treated with rapamycin. Pre-existing higher basal levels of mTORC1 signaling in shAMPK cells could suppress the maximum amount AKT can be stimulated in these cells.

Each experiment currently shown in Sup Fig. 8 should include the following:

- a. Rapamycin treatments (needed for experiments with EGF, Glucose deprivation, and H₂O₂ - Sup Fig. 8b-d).
- b. Readouts for Akt signaling including Phospho-T308-Akt, phospho-S473-Akt, Total Akt, phospho-PRAS40, and total PRAS40 (need to be shown for EGF, glucose deprivation, and H₂O₂ experiments - Sup Fig 8b-d).
- c. Readouts for AMPK signaling including total levels of AMPK to confirm level of knockdown in each experiment and Phospho-ACC and total ACC (needed in Sup FigA-D).

AUTHOR RESPONSE: We thank the reviewer for the thoughtful inputs. We agree that our statement may be too strong and it's better not to totally rule out AMPK's function on mTORC1-dependent negative feedback of Akt activation. We have followed the reviewer's suggestion to modify the statement as well as adding additional rapamycin treatment experiments. For the data on mTORC1, we have moved them to the Result section according to the reviewer's suggestion. The revised statement and figures are shown in **page 8** and **revised Supplementary Fig. 3a-d**. Control and AMPK knockdown cells were treated with rapamycin under hypoxia, glucose deprivation, H₂O₂ or EGF. While phosphorylation of mTOR downstream effectors, S6K and Grb10, was induced under these conditions, AMPK knockdown has subtle impact on phosphorylation of S6K and Grb10 in our cell models. However, AMPK knockdown consistently impaired Akt phosphorylation even under the treatment of mTORC1 inhibitor rapamycin (**Supplemental Fig. 3a-d**). As a control, we found that mTORC1 inhibitor rapamycin inhibited S6K and Grb10 phosphorylation. These data suggest that AMPK dependent Akt activation under hypoxia, glucose deprivation, H₂O₂ and EGF partly acts through an mTORC1-independent mechanism.

Moreover, we followed the reviewer's suggestions by including the similar readouts for Akt signaling including phospho-T308-Akt, phospho-S473-Akt, total Akt, phospho-PRAS40, total PRAS40 as well as readouts for AMPK signaling including total AMPK level to confirm the level of knockdown in each experiment, and phospho-ACC and total ACC level to monitor AMPK activation (**Supplemental Fig. 3a-d**).

2) A readout of Akt signaling (phospho-PRAS40 and total PRAS40) is missing from Fig 1C and 1D. Why wouldn't you make Figures 1C, 1D, and 1E all uniform with the same blots in the same order?? Phospho-PRAS40 is also missing from Fig 2A, 2C, 2D, and 2F, while it is shown in 2B and 2E. Show it in every figure so that they have a readout in common! You can't just show phospho-GSK3 in some subfigures and phospho-PRAS40 in others for no good reason. Be consistent. There's other figures where only phospho-Akt or phospho-GSK3 is shown but I'm not going to list them all. Again, bottom line is there should be the same readouts shown consistently across figures so readers can properly compare across different cell lines, treatments, and types of experiments.

AUTHOR RESPONSE: We apologized to the reviewer for our inconsistent use of Akt substrates. We have followed the reviewer's suggestion to fill the missing blots for PRAS40 phosphorylation in **Fig.1C, 1D; Fig 2A, C, D, F; Supplementary Fig. 3a-d; Supplementary Fig. 5b-i**. PRAS40 phosphorylation is consistent with Akt T308 phosphorylation in multiple cell lines under the treatment of diverse stresses and EGF.

3) There's no description of how AMPK/ CaMKKB Addback cells were made.

AUTHOR RESPONSE: The information was described in the revised method section.

4) Need to show EGFR and ERK phosphorylation in the only addback experiment in this figure, Fig. 2B.

AUTHOR RESPONSE: EGFR and ERK phosphorylation was shown in the **revised Fig. 2B**.

5) If shCAMKKb and shLKB1 knockdowns are going to be shown in separate experiments then at the very least show the exact same blots/readouts. For instance, why would you show phospho-ACC in Fig 2C but phospho-AMPK in SupFig 2E? Why isn't a readout for Akt signaling shown in SupFig 2E? Why aren't p-EGFR and pERK shown in SupFig 2E? Just show the same exact set of blots for Fig2C and SupFig 2E.

AUTHOR RESPONSE: Experiments using shCAMKKb and shLKB1 knockdown cells were repeated to present the same readouts in **revised Fig. 2C** and **Supplementary Fig. 2e**.

6) need to describe the AMPKa1b1g1 complex used here better in methods. Where did it come from? How was it purified?

AUTHOR RESPONSE: Recombinant AMPKa1b1g1 complex, which was purified from Baculovirus infected Sf9 cells, was purchased from abcam (ab79803). This information and the Category number were updated in revised method section (Recombinant Protein Purification and in vitro kinase assay).

7)CA is expressed ~4X higher (molar) or more than KD? Not to mention a full length KD construct is being compared to a truncated CA construct. Not the best control.

AUTHOR RESPONSE: We agree that CA is not the best control to compare with KD but it is an acceptable tool to study the function of activated AMPK. In addition, we also restored WT and KD AMPK at the same level in AMPK null MEFs in more relevant, physiological conditions (**Fig. 1c-e**) and found that WT AMPK, but not AMPK KD, rescued the defect in Akt T308 phosphorylation under diverse stresses.

8) there seems to be a disconnect between Skp2 phosphorylation and Akt phosphorylation in two ways. First, I would point out that in response to H₂O₂ the increase in phospho-Akt (at 1h) precedes any increase in phospho-Skp2 (at 2h) by a whole hour. How could Skp2 phosphorylation mediate the increase in phospho-Akt if Akt phosphorylation precedes Skp2 phosphorylation? Doesn't make sense. On the flipside of this point, for EGF and glucose deprivation, phospho-Skp2 decreases at later time points while AMPK signaling and phospho-Akt are still going up (for EGF, Fig 2A vs. Fig 4G; for GD, Fig 1G vs. Fig 4G). Again, it's hard to see how Skp2 phosphorylation is mediating Akt phosphorylation when they are going in opposite directions. I can't tell if something similar is happening with hypoxia because an 8h time point isn't shown in Fig 1B.

AUTHOR RESPONSE: We thank the reviewer for pointing out the discrepancy in kinetics between Skp2 S256 phosphorylation and Akt T308 phosphorylation under H₂O₂ treatment. To resolve this issue, we have re-run the lysates and showed that Skp2 S256 phosphorylation could be induced at 1 hour of H₂O₂ treatment and reached to the higher level after 2 hours of H₂O₂ treatment in the **revised Fig.4g**, consistent with the result for Akt T308 phosphorylation upon H₂O₂ treatment in **Fig. 1d**.

For the second question, the reviewer is puzzling about the fact that Skp2 S256 phosphorylation dropped at a later time but Akt activation is still sustained. Our earlier data showed Akt K63-linked ubiquitination induced by Skp2 SCF complex promotes the cell membrane localization of Akt and subsequent Akt T308 phosphorylation by PDK1. In this study, we show that AMPK activation phosphorylates Skp2 at S256 is critical for Skp2 SCF complex formation and its E3 ligase activity, thereby leading to Akt ubiquitination and subsequent Akt T308 phosphorylation. It is possible that although Skp2 S256 phosphorylation, which is an early and critical event for Akt T308 phosphorylation under stress, did not sustain for a long period of time under stress, it is sufficient to drive Akt ubiquitination and subsequent Akt T308 phosphorylation.

MINOR COMMENTS

1) Recheck labeling on all figures.

AUTHOR RESPONSE: Misalignments of labels were carefully checked and fixed.

2) "not significant" should be reserved for referring to statistical significance.

AUTHOR RESPONSE: Text changed to “not statistical significant” in the revised manuscript.

3) Please include the manufacturers catalogue number for reagents when applicable in methods section including for antibodies. This is standard practice these days.

AUTHOR RESPONSE: Antibodies and reagents were updated with catalogue number.

4) For some “phospho-” antibodies in methods it says “phosphor”. Please change all to “phospho-” in methods and throughout text for consistency.

AUTHOR RESPONSE: The corrections were made accordingly in the method under antibody section and throughout text.

5) Clearly label and state in legend for ALL main and supp figures which specific Skp2 and AMPKa shRNAs are being used since there are two for each. For instance, which AMPK shRNA is used in Fig 1b (shAMPKa1-1 or 1-2?)

AUTHOR RESPONSE: Specific shRNA clones were updated in revised manuscript and figures.

6) The title of the paper doesn't fully reflect the contents of the paper – there's no mention of stress induced Akt activation.

AUTHOR RESPONSE: Thanks for reviewer's suggestion. The title is changed to “The critical role of stress kinase AMPK in driving oncogenic Akt activation under stress and EGF, tumorigenesis and drug resistance”.

REVIEWERS' COMMENTS:

Reviewer #3 (Remarks to the Author):

The manuscript has much improved and the authors have addressed my major concerns.

But, one issue that I'd like to point out is that in response to one of my comments on Fig 4G, the authors ran the same lysates and got a different looking blot that now conforms better to the expected result (see attached file and point 6 below). This seems strange. I think the authors should explain to the editor how this is possible.

Here are a few final minor comments.

1) I know this is a bit presumptuous but might I suggest slightly different wording for the title to make it flow better: "The critical role of the stress kinase AMPK in driving oncogenic Akt activation under conditions of stress, EGF stimulation, tumorigenesis, and drug resistance." Take it or leave it. I know it's unusual to make such a suggestion.

2) Page 7, Line 186. "statistical" should be "statistically"

3) Page 8, Line 221. Should add that AMPK also phosphorylates Raptor within mTORC1 (along with TSC2) to inhibit mTORC1

4) Fig 2B. For clarity label the wild type MEFs "AMPK α +/+" instead of just "WT" in both the legend and Fig 2B.

5) Methods. There are still some reagents that should have catalogue numbers listed and not just the company including EGF, drugs, matrigel, cell counting kit, glucose-free DMEM. All but the most basic reagents should include catalogue numbers. There can be different versions or formulations from the same company.

6) Fig 4G, H₂O₂ blot. The kinetics of p-Skp2 now make more sense relative to p-Akt in other blots. HOWEVER, I have to say that I am confused as to how the authors ran the SAME lysates and got a quite different looking blot for p-Skp2 that happens to now correlate better with p-Akt?? (see attached file for comparison of old vs new versions)

Why do the bands in the shLUC 0h and 1h timepoint look different now when they were equal in the original? Why did the bands in the shAMPK lanes disappear in the new version when they were equal with the 0h and 1h bands in the shLUC in the original? To be honest, this type of magical change inevitably raises suspicions.

Reviewer #3 (Remarks to the Author):

The manuscript has much improved and the authors have addressed my major concerns.

But, one issue that I'd like to point out is that in response to one of my comments on Fig 4G, the authors ran the same lysates and got a different looking blot that now conforms better to the expected result (see attached file and point 6 below). This seems strange. I think the authors should explain to the editor how this is possible.

Here are a few final minor comments.

1) I know this is a bit presumptuous but might I suggest slightly different wording for the title to make it flow better: "The critical role of the stress kinase AMPK in driving oncogenic Akt activation under conditions of stress, EGF stimulation, tumorigenesis, and drug resistance." Take it or leave it. I know it's unusual to make such a suggestion.

Response: Thanks for the suggestion. However, we have to shorten the title to meet Nature Communications' requirement, which is only 15 words allowed for the title.

2) Page 7, Line 186. "statistical" should be "statistically"

Response: We thank reviewer for checking carefully for our manuscript. We have now made the change accordingly.

3) Page 8, Line 221. Should add that AMPK also phosphorylates Raptor within mTORC1 (along with TSC2) to inhibit mTORC1

Response: We followed reviewer's suggestion to add Raptor to the indicated text.

4) Fig 2B. For clarity label the wild type MEFs "AMPK α +/+" instead of just "WT" in both the legend and Fig 2B.

Response: We corrected WT to AMPK α +/+ in Figure 2B and legend.

5) Methods. There are still some reagents that should have catalogue numbers listed and not just the company including EGF, drugs, matrigel, cell counting kit, glucose-free DMEM. All but the most basic reagents should include catalogue numbers. There can be different versions or formulations from the same company.

Response: We have added the catalogue number for EGF (R&D, Cat# 236-EG-200), Kinase inhibitors (Selleckchem, Cat# L1200), matrigel (Corning, Cat# 354234), cell counting kit (Vita Scientific, Cat# CK04-05), glucose-free DMEM (Gibco, Cat# 11966025).

6) Fig 4G, H₂O₂ blot. The kinetics of p-Skp2 now make more sense relative to p-Akt in other blots. HOWEVER, I have to say that I am confused as to how the authors ran the SAME lysates and got a quite different looking blot for p-Skp2 that happens to now correlate better with p-Akt?? (see attached file for comparison of old vs new versions)

Why do the bands in the shLUC 0h and 1h timepoint look different now when they were equal in the original? Why did the bands in the shAMPK lanes disappear in the new version when they were equal with the 0h and 1h bands in the shLUC in the original? To be honest, this type of magical change inevitably raises suspicions.

Response: We apologize that we forget to make a clear statement of the difference between the original and the repeated experiment. We indeed re-purified our phospho-Skp2 S256 antibody using the Skp2 S256 phospho-peptide. Specifically, we bind the Skp2 S256 phospho-peptide to CNBr-activated sepharose 4-B and follow the manufacturer's instruction to purified the antibody (Sigma, Cat# GE17-0430-01). After re-purification of the antibody, it's more sensitive to detect the change in 1h H₂O₂ treatment and reduced the basal signal in AMPK knockdown cells.